# Neuronal sFlt1 and Vegfaa determine venous sprouting and spinal cord vascularization

Raphael Wild[1,2], Alina Klems[1,2], Masanari Takamiya[2], Yuya Hayashi[2,3], Uwe Strähle[2], Koji Ando[4], Naoki Mochizuki[4], Andreas van Impel[5,6], Stefan Schulte-Merker[5,6], Janna Krueger[7], Laetitia Preau[1] & Ferdinand le Noble[1,2]

Formation of organ-specific vasculatures requires cross-talk between developing tissue and specialized endothelial cells. Here we show how developing zebrafish spinal cord neurons coordinate vessel growth through balancing of neuron-derived Vegfaa, with neuronal sFlt1 restricting Vegfaa-Kdrl mediated angiogenesis at the neurovascular interface. Neuron-specific loss of *flt1* or increased neuronal *vegfaa* expression promotes angiogenesis and peri-neural tube vascular network formation. Combining loss of neuronal *flt1* with gain of *vegfaa* promotes sprout invasion into the neural tube. On loss of neuronal *flt1*, ectopic sprouts emanate from veins involving special angiogenic cell behaviours including nuclear positioning and a molecular signature distinct from primary arterial or secondary venous sprouting. Manipulation of arteriovenous identity or Notch signalling established that ectopic sprouting in *flt1* mutants requires venous endothelium. Conceptually, our data suggest that spinal cord vascularization proceeds from veins involving two-tiered regulation of neuronal sFlt1 and Vegfaa via a novel sprouting mode.

[1] Department of Cell and Developmental Biology, Institute of Zoology (ZOO) Karlsruhe Institute of Technology (KIT), Fritz-Haber-Weg 4, 76131 Karlsruhe, Germany. [2] Institute for Toxicology and Genetics (ITG), Karlsruhe Institute of Technology (KIT), PO Box 3640, 76021 Karlsruhe, Germany. [3] Department of Molecular Biology and Genetics, Aarhus University, Gustav Wieds Vej 10, 8000 Aarhus C, Denmark. [4] Department of Cell Biology, National Cerebral and Cardiovascular Research Institute, 5-7-1 Fujisirodai, Suita, Osaka 565-8565, Japan. [5] Institute for Cardiovascular Organogenesis and Regeneration, Faculty of Medicine, University of Münster, Mendelstr. 7, 48149 Münster, Germany. [6] Cells-in-Motion Cluster of Excellence, (EXC 1003-CiM), University of Münster, Waldeyerstraße 15, 48149 Münster, Germany. [7] Department of Translational Oncology, Biological Sciences Platform, Sunnybrook Research Institute, 2075 Bayview Ave., M4N 3M5 Toronto, Canada. Correspondence and requests for materials should be addressed to F.l.N. (email: ferdinand.noble@kit.edu).

The vascular network closely associates with the neuronal network throughout embryonic development, in adulthood and during tissue regeneration[1–3]. Close association of vessels and nerves allows reciprocal cross-talk involving diffusible molecules, which is important for physiological functions in both domains[4,5]. Arteries secrete factors that attract sympathetic axons, and adrenergic innervation of arteries allows the autonomic nervous system to control arterial tone and tissue perfusion[5]. The nervous system, on the other hand, requires a specialized network of blood vessels for its development and survival. Metabolically active nerves rely on blood vessels to provide oxygen necessary for sustaining neuronal activity[6], and disturbances herein result in neuronal dysfunction[1,7].

How nerves attract blood vessels is debated, but several studies addressing vascularization of the mouse and chicken embryonic nervous system suggest that the angiogenic cytokine VEGF-A is involved[8–10]. In the mouse peripheral nervous system axons of sensory nerves innervating the embryonic skin trigger arteriogenesis involving VEGF-A–Neuropilin-1 (NRP1) dependent signalling[11,12]. While these studies provide evidence for the physical proximity and cooperative patterning of the developing nerves and vasculature, relatively little is known about mechanisms controlling VEGF-A dosage at the neurovascular interface. This is of great importance considering that blood vessels are very sensitive to changes in VEGF-A protein dosage and even moderate deviations from its exquisitely controlled physiological levels result in dramatic perturbations of vascular development[13,14]. VEGF-A levels must therefore be well titrated, and several strategies have evolved to achieve this.

Mouse retinal neurons for example can reduce extracellular VEGF-A protein via selective endocytosis of VEGF-A–VEGF receptor-2 (KDR/FLK) complexes. Inactivation of this uptake causes non-productive angiogenesis[15]. In the vascular system, spatio-temporal control of VEGF-A protein dosage is thought to be achieved by soluble VEGF receptor-1 (sFLT1), an alternatively spliced, secreted isoform of the cell-surface receptor membrane-bound FLT1 (mFLT1)[16,17]. Soluble FLT1 binds VEGF-A with substantially higher affinity than KDR, thereby reducing VEGF-A bioavailability and attenuating KDR signalling[17]. While originally discovered as a vascular-specific receptor, evidence is emerging showing neuronal FLT1 expression[18]. To what extent endogenous neuronal Flt1 has a physiological role in titrating neuronal VEGF levels controlling angiogenesis at the neuro-vascular interface independent of vascular Flt1 remains to be determined.

Angiogenesis involves complex and dynamic changes in endothelial cell behaviour[19]. In the zebrafish embryo these events can be studied in detail at the single cell level *in vivo* through the use of vascular-specific reporter lines[20,21]. The stereotyped patterning of arteries and veins in the trunk of the zebrafish embryo prior to 48 hpf is mediated by cues derived from developing somites and the hypochord, controlling angiogenic sprout differentiation and guidance[22,23]. Sprouting of intersegmental arterioles (aISV) requires Vegfaa-Kdrl signalling, as loss of either *kdrl* or *vegfaa* completely abolishes ISV sprouting from the dorsal aorta (DA)[24]. Primary sprouting also involves a component regulated by Notch, as loss of Notch increases the endothelial propensity to occupy the tip cell position in this vessel, whereas gain of Notch restricts aISV development[25]. Secondary vein sprouting requires Vegfc-Flt4 signalling, as loss of either ligand or receptor blocks venous growth[26,27]. Developing somites are regarded as the main source for Vegfaa, while the hypochord provides Vegfc during early development[22,23].

In this study we show that developing spinal cord neurons located in the trunk of the zebrafish embryo produce Vegfaa and sFlt1 affecting the angiogenic behaviour of intersegmental vessels at the neurovascular interface. We find that during early development neuronal sFlt1 restricts angiogenesis around the spinal cord. We demonstrate that on genetic ablation of neuronal sFlt1 this brake is relieved resulting in the formation of a vascular network supplying the spinal cord in a Vegfaa-Kdrl dependent manner. Using inducible neuron-specific *vegfaa* gain-of-function approaches and analysis of several mutants with *vegfaa* gain-of-function scenarios, we furthermore show that the neuronal Vegfaa dosage determines the extent of the neovasculature supplying the spinal cord, as well as sprout invasion into the spinal cord. Interestingly, loss of *flt1* or augmenting neuronal *vegfaa* promotes sprouting from intersegmental veins involving distinctive angiogenic cell behaviours including nuclear positioning and a molecular signature not observed in primary arterial or secondary venous sprouting. Cell transplantation experiments confirm the role of neuronal *flt1* in venous sprouting and furthermore show that vascular *flt1* is dispensable herein. Taken together, our data suggest that spinal cord vascularization proceeds from veins and is coordinated by two-tiered regulation of neuronal sFlt1 and Vegfaa determining the onset and the extent of the vascular network that supplies the spinal cord via a novel sprouting mode.

## Results

**Spinal cord neurons express *sflt1*, *mflt1* and *vegf* ligands.** Analysis of *TgBAC(flt1:YFP)*[hu4624]*;Tg(kdrl:hsa.HRAS-mCherry)*[s916] transgenic embryos showed *flt1* expression in the aorta, arterial intersegmental vessels (aISVs), dorsal part of venous intersegmental vessels (vISVs) and spinal cord neurons located in the neural tube (Fig. 1a,b,d–g)[18]. Spinal cord neurons were in close proximity to blood vessels (Fig. 1c–e) and 3D-rendering of confocal z-stacks obtained from *Tg(kdrl:EGFP)*[s843]*;Tg(Xla.Tubb:DsRed)*[zf148] double transgenic embryos showed the dorsal aspect of ISVs 'indenting' the neural tube indicative of close contact (Fig. 1c; Supplementary Movie 1). Optical sections confirmed close contact between the outer neuronal layers of the neural tube and the dorsal part of ISVs, as well as the dorsal longitudinal anastomotic vessel (DLAV) (Fig. 1d–g). Such anatomical juxtapositioning of trunk vessels and neurons may provide a template for molecular cross-talk (Fig. 1d,e; pink box).

TaqMan analysis using FAC-sorted neuronal cells from two different neuronal reporter lines (Supplementary Fig. 1a–k) showed expression of *mflt1*, *sflt1*, *kdrl*, *kdr*, *flt4* and the ligands *vegfaa*, *vegfab*, and *plgf* (Supplementary Fig. 1b,e)[24]. *Flt1* was expressed in a comparable range as neuronal guidance molecules (Supplementary Fig. 1c,f). Real-time qPCR analysis for *vegfaa* and *vegfab* in the trunk of developing zebrafish embryos confirmed expression of both isoforms (Supplementary Fig. 1l,m).

**Loss of *flt1* induces ectopic vascular networks.** In zebrafish *flt1* consists of 34 exons encoding membrane-bound *mflt1* and soluble *sflt1*, which is formed by alternative splicing at the exon 10—intron 10 boundary (Supplementary Fig. 2a)[18]. To obtain loss of both *mflt1* and *sflt1* (*flt1* full mutants) we targeted *flt1*-exon 3 using a CRISPR/Cas approach (Supplementary Fig. 2a–d) and analysed in detail the vascular phenotypes of three mutant alleles, *flt1*[ka601] (−1 nt), *flt1*[ka602] (−5 nt) and *flt1*[ka603] (+5 nt) (Fig. 2a–f; Supplementary Fig. 2a–d). To obtain *mflt1*-specific mutants we targeted exon 11b, the alternative exon essential for *mflt1* transcription (Fig. 2g; Supplementary Fig. 2a,e)[18]. Both the *flt1*[ka601] and *flt1*[ka605] mutant showed no signs of non-sense mediated decay (Supplementary Fig. 3a,b).

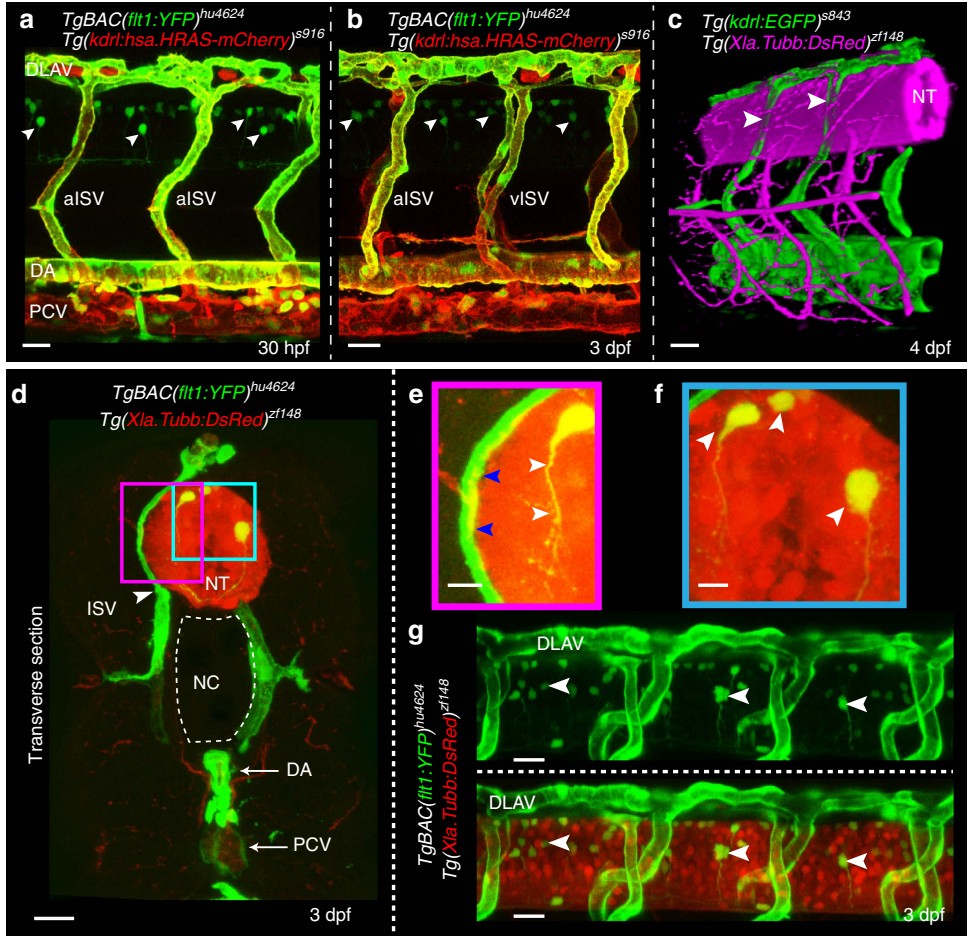

**Figure 1 | Expression of Vegf receptors and ligands at the neurovascular interface.** (**a**,**b**) Double transgenic embryos *TgBAC(flt1:YFP)*^hu4624; *Tg(kdrl:hsa.HRAS-mcherry)*^916 at 30 hpf and 3 dpf shows *flt1* expression (green) in dorsal aorta, arterial ISV and dorsal aspect of venous ISV (3 dpf) and neurons (arrowheads). (**c**) 3D-rendered view of vessels (green) and nerves (purple) in *Tg(kdrl:EGFP)*^s843;*Tg(Xla.Tubb:DsRed)*^zf148 embryos highlighting dorsal aspect of ISVs (arrowheads) in close contact to the neural tube (NT). (**d**) Transverse section of the trunk of *TgBAC(flt1:YFP)*^hu4624; *Tg(Xla.Tubb:DsRed)*^zf148 embryos shows that ISVs (green, arrowhead) and neural tube (NT, red) are in close contact. Dorsal is up. (**e**) Magnified view of purple-boxed area in (**d**), showing direct contact of vessels with nerves at the neurovascular interface (blue arrowheads) and *flt1* expressing neurons with long axonal extensions in the neural tube (white arrowheads). (**f**) Magnified view of blue-boxed area in (**d**) showing *flt1* expressing neurons (arrowheads) and their axons inside neural tube (red). (**g**) Lateral view of *TgBAC(flt1:YFP)*^hu4624; *Tg(Xla.Tubb:DsRed)*^zf148 at the level of the neural tube showing *flt1* expressing neurons (arrowheads) in neural tube. DA, dorsal aorta; dpf, days post fertilization; DLAV, dorsal longitudinal anastomotic vessel; hpf, hours post fertilization; ISV, intersegmental vessel; NC, notochord; NT, neural tube; PCV, posterior cardinal vein. Scale bar, 30 μm in **a–d**,**g**; 10 μm in **e**,**f**.

Zebrafish homozygous for the *flt1* -1 nt allele (*flt1*^ka601) displayed severe hyper-branching of the trunk vasculature at 3–4 dpf (Fig. 2a,b). Supernumerous amounts of branches developed in the dorsal aspect of the trunk at the level of the neural tube (Fig. 2b,c). Comparable observations were made in embryos homozygous for the *flt1* − 5 nt allele (*flt1*^ka602) and the *flt1* + 5 nt allele (*flt1*^ka603) (Fig. 2d,e,f). Analysis of four *mflt1* mutant alleles (*flt1*^ka605-608, Supplementary Fig. 2e) did not reveal any obvious vascular malformations or alterations in vascular branching morphogenesis (Fig. 2g). These observations are compatible with absence of angiogenic defects in mouse *Flt1*^TK−/− embryos lacking mFlt1 signalling[28,29]. The vascular phenotype observed in the *flt1*^ka601 mutants thus most likely involved soluble Flt1.

Since the vascular phenotypes of the *flt1*^ka601, *flt1*^ka602 and *flt1*^ka603 mutant alleles (*flt1* full mutants) were indistinguishable, we focused on analysing *flt1*^ka601 embryos (Fig. 2p–s). *Flt*^ka601 mutants showed normal arterial-venous remodelling (Fig. 2b,p–s) and adequate perfusion of both aISVs and vISVs.

No significant changes in heart frequency were noted (Supplementary Fig. 4a). The vascular phenotype of *flt1*^ka601 mutants emerged around day 2.5 (Fig. 2q,r) with sprouts emanating exclusively from the dorsal aspect of the venous ISVs at the level of the neural tube (Supplementary Movie 2); ectopic arterial ISV sprouting was not observed (Fig. 2r). In *flt1*^ka601/+ heterozygotes (Fig. 2l–o) ectopic sprouting was rarely observed (Fig. 2n,o; Supplementary Movie 3). In wild-type (WT) embryos such endothelial cell behaviours were not observed (Fig. 2h–k, Supplementary Movie 4).

We furthermore examined whether *flt1* targeting morpholino could recapitulate the *flt1*^ka601 mutant phenotype (Supplementary Fig. 4b–h). We evaluated two dosages of a published *flt1* ATG targeting morpholino (MO) and found that 1 ng *flt1* MO induced hyper-branching in WT at levels comparable to *flt1*^ka601 (Supplementary Fig. 4g,h)[18,30]. Injection of 1 ng MO into *flt1*^ka601 mutant background did not induce additional sprouting defects (Supplementary Fig. 4d), suggesting that the 1ng dosage targets *flt1* specifically. In contrast, 3 ng MO introduced

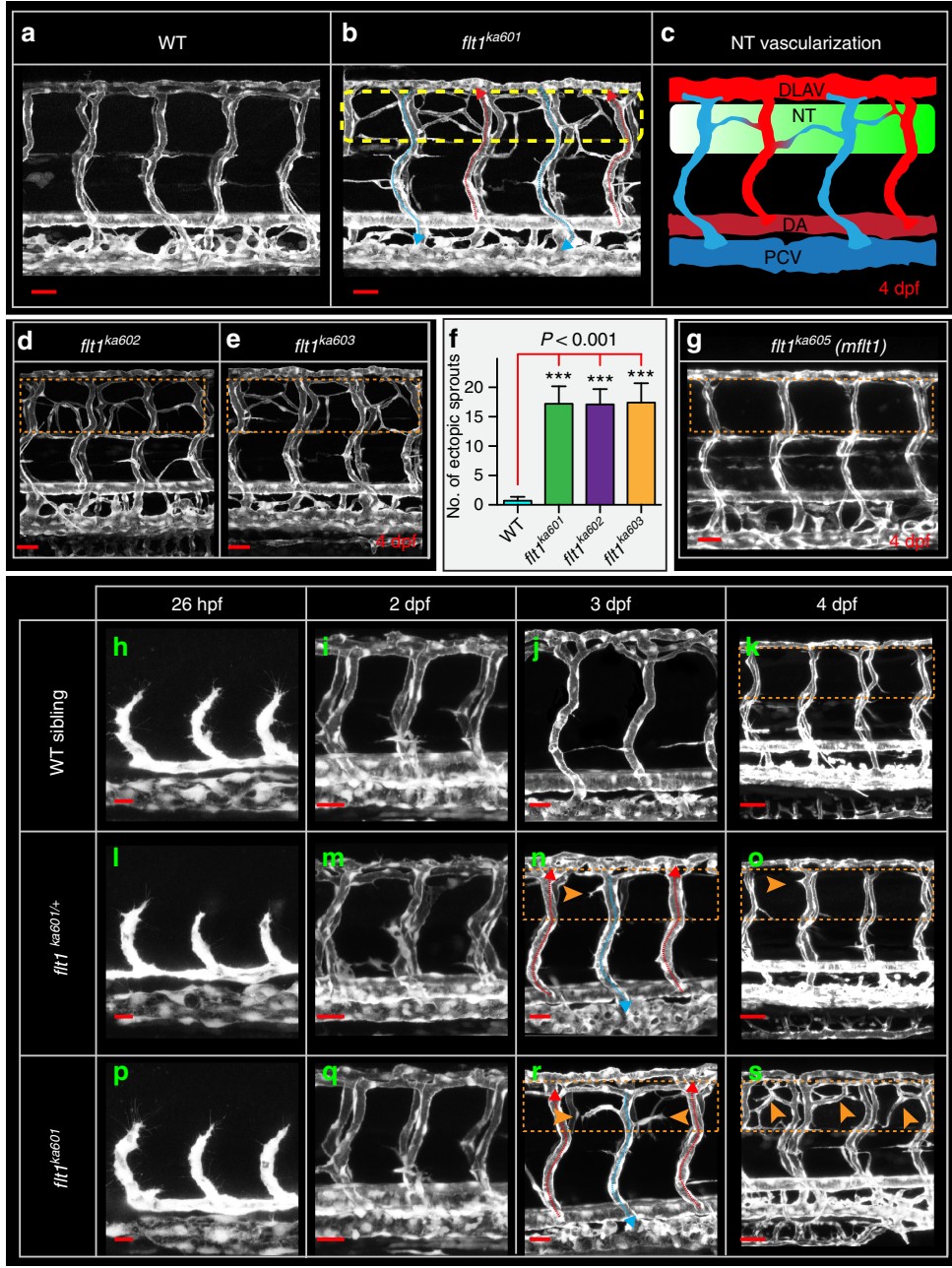

**Figure 2 | *Flt1* mutants develop hyper-branched vascular networks at the level of the neural tube.** (**a**) Trunk vasculature in 4 dpf WT sibling, (**b**) trunk vasculature in 4 dpf *flt1^ka601^* mutant, in *Tg(kdrl:EGFP)^s843^* background. Perfused aISVs with red arrow, veins with blue arrow. Note the extensive amount of hyper-branching (dotted box) at the level of the neural tube. (**c**) Schematic representation of hyper-branching phenotype along the neural tube; ectopic vessels make anastomosis between vISV (blue) with aISVs (red). (**d**) Hyper-branching (dotted box) is also observed in *flt1^ka602^* and (**e**) *flt1^ka603^* mutants. (**f**) Quantification of hyper-branching for indicated mutant alleles. Mean ± s.e.m., n = 10 per group, ANOVA. (**g**) Membrane-bound *flt1* mutant (*flt1^ka605^*) without vascular phenotype (compare dotted box in **g**, with control in **a**). (**h–k**) Trunk vascular network in WT embryos at indicated time points. (**l–o**) Trunk vasculature in *flt1^ka601/+^* embryos at indicated time points. (**p–s**) Trunk vasculature in *flt1^ka601^* embryos at indicated time points. Arrowheads indicate ectopic branches. DA, dorsal aorta; PCV, posterior cardinal vein; DLAV, dorsal longitudinal anastomotic vessel; NT, neural tube; hpf, hours post fertilization; dpf, days post fertilization. Scale bar, 50 μm in **a,b,d,e,g,i,m,q,k,o,s**; 25 μm in **h,l,p,j,n,r**.

additional branches at 2 dpf that were not observed in the *flt1^ka601^* mutant at this stage (Supplementary Fig. 4e). Since we did not observe maternal contribution of *flt1* these observations suggest that 3 ng MO introduced non-specific effects[31].

**Sprouts in *flt1^ka601^* display distinctive cell behaviours.** Compatible with ectopic sprouting we identified hyperactive

endothelial cells extending filopodia in the dorsal aspect of vISVs of *flt1^ka601^* mutants (Fig. 3a; Supplementary Movie 2). About 55% of hyperactive endothelial cells investigated generated a patent sprout (Fig. 3a); in the remaining 45%, filopodia and sprouts retracted (Fig. 3b). From the population of patent ectopic venous sprouts 95% formed an anastomosis with an aISV, whereas only 5% made a connection with a vISV (Fig. 3c). The preference for arterial anastomosis may be physiologically

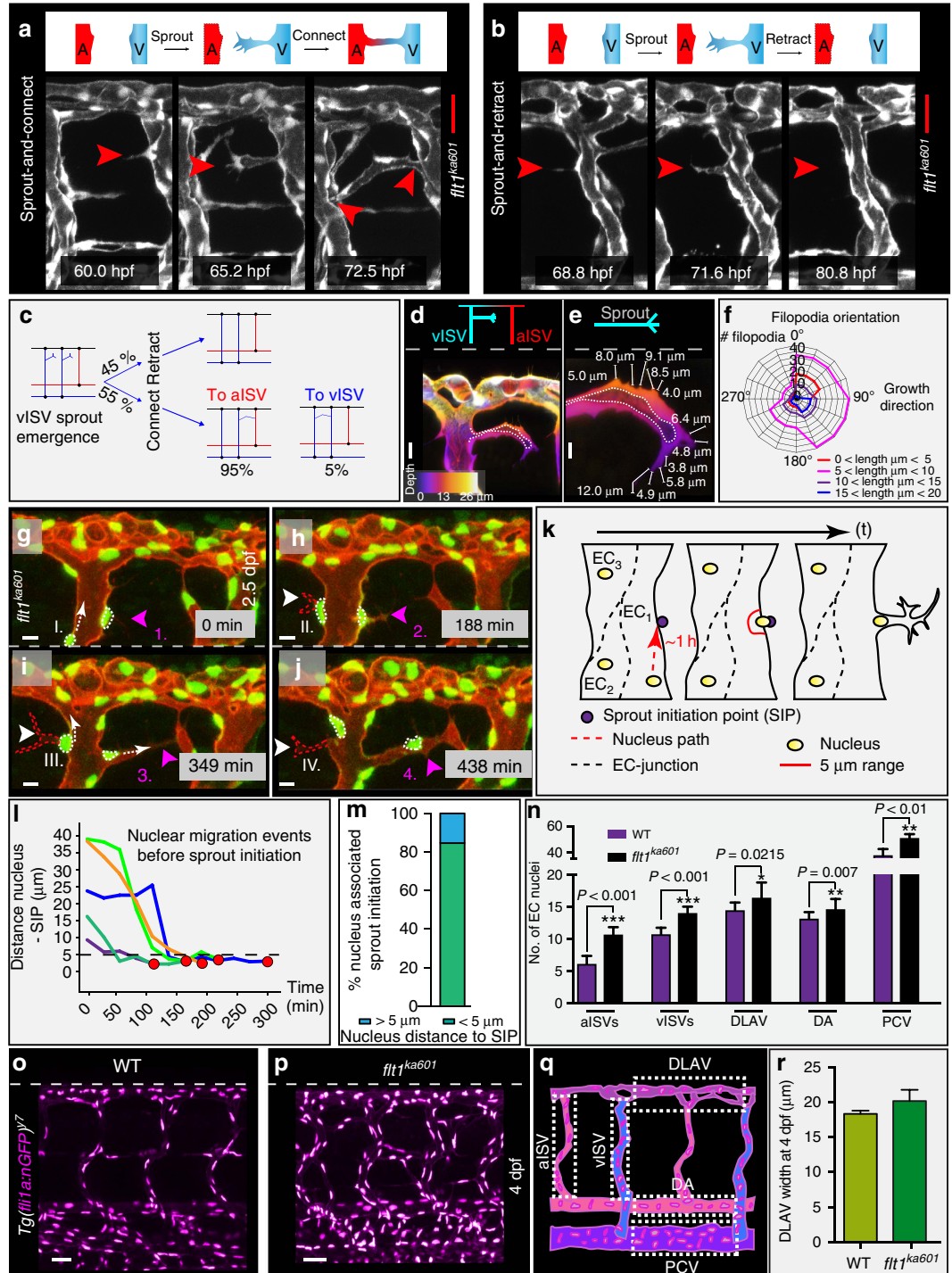

**Figure 3 | Imaging and quantification of sprouting kinetics in *flt1^ka601* mutants.** (**a**) Time lapse imaging of sprout initiation and anastomosis formation in *flt1^ka601* mutant. Sprout initiation (60.0 hpf), elongation (65.2 hpf) and connection-anastomoses (72.5 hpf) with adjacent aISV. (**b**) Time lapse imaging of sprout initiation and retraction in *flt1^ka601* mutant. Endothelial cells produce filopodia (68.8 hpf), extend a sprout (71.6 hpf), which subsequently retracts (80.8 hpf). (**c**) Quantification of data in (**a,b**) showing % of sprouts retracting (top part, 45%) or connecting (bottom part, 55%) to adjacent ISVs. In the latter scenario sprouts in 95% of cases connected to aISV (red) and in 5% of cases to vISV (blue). Angiogenic behaviour was analysed in time-lapse confocal movies, *n* = 20 embryos. (**d–f**) Filopodia directionality and length in *flt1^ka601* mutants (*n* = 10 embryos, *n* = 920 filopodia). (**g–j**) Time lapse imaging of endothelial nuclei in *Tg(fli1a:nGFP)^y7; Tg(kdrl:hsa.HRAS-mcherry)^s916* showing association between nuclear position and sprouting initiation point (SIP). Note that sprouts arise in close proximity to the position of the nucleus. Arrowheads indicate sprouts; nuclei at indicated time points (sprout initiation with actively migrating nucleus towards SIP I, II, III, IV and nucleus already located at SIP 1,2,3,4). (**k**) Schematic representation of nuclear position with respect to SIP. (**l,m**) Quantification of observations in **g–j**. Red dot indicates sprout initiation time point. Note that sprouting preferentially occurs when endothelial nuclei are within less than 5 μm from SIP (SIP below dotted line in (**l**)). *n* = 5 (**l**) and *n* = 13 (**m**). (**n–q**) Quantification of EC nuclei in aISV, vISVs, DLAV, DA and PCV of WT and *flt1^ka601* embryos at 4 dpf; mean ± s.e.m., *t*-test, *n* = 21 embryos per genotype. (**r**) Quantification of DLAV width in WT and *flt1^ka601* mutant, *n* = 9 embryos per genotype. A, artery; aISV, arterial intersegmental vessel; DA, dorsal aorta; DLAV, dorsal longitudinal anastomotic vessel; EC, endothelial cell; PCV, posterior cardinal vein; SIP, sprout initiation point; V, vein; vISV, venous intersegmental vessel. Scale bar, 30 μm in **a,b**; 10 μm in **d,e,g–j**; 50 μm in **o,p**.

relevant as it creates a pressure gradient promoting blood flow perfusion. Sprout filopodia length ranged from 1 to 20 μm, and filopodia projected at an angle between 90 and 120° with respect to the vISV compatible with arterial anastomosis formation (Fig. 3d–f). Current models posit that Flt1 produced in angiogenic sprouts mainly prevents back-branching of nascent sprouts[32]. We find that in the absence of Flt1 sprouts retain their directionality and migrate away from the parent vessel. Within ISVs endothelial nuclei migrated at velocities of up to 1 μm min$^{-1}$ (Fig. 3g–j,l). Careful analysis of nuclear positioning within endothelial cells revealed an association between nuclear position and sprout initiation (Fig. 3k–m, Supplementary Movie 2). Nuclei migrated actively into the direction of future sprout initiation points (SIP), and in more than 80% of the studied sprout initiations nuclear positioning was directly linked with sprout initiation (linkage was defined as nucleus-SIP distance of <5 μm at spout initiation) (Fig. 3l,m). This nuclear migration behaviour is in contrast to rearward nuclear positioning in migrating angiogenic endothelial cells in vitro[33] and is not described in vivo for primary artery or secondary venous sprouting events in zebrafish. Analysis of endothelial cell numbers at 4 dpf showed increased endothelial cell numbers in aISVs, vISVs, DLAV, DA and PCV of $flt1^{ka601}$ compared with WT (Fig. 3n–q); DLAV size was not statistically different (Fig. 3r). At earlier stages (17 hpf) we found no differences in endothelial cell numbers between $flt1^{ka601}$ and WT (Supplementary Fig. 4i–l).

**$flt1^{ka601}$ display upregulation of angiogenic sprout markers.** We next performed RNA sequencing of $flt1^{ka601}$ mutants and analysed genes implicated in sprouting angiogenesis (Supplementary Fig. 5a–c). Expression of the classical tip-stalk cell markers including notch1a, notch1b, dll4, nrarpa, nrapb, hey1, hey2, her6 and flt4 were not altered[34] (Supplementary Fig. 5b). This result may not be surprising since ectopic venous sprouts emanated from venous ISVs, and Dll4-Notch signalling is absent in this domain[35]. Instead we found upregulated expression of other genes implied in sprouting cell behaviour. RNA-seq and qPCR of flt1 mutants showed significantly increased levels of apelin receptor-a (aplnra), angiopoietin-2a (angpt2a), and endothelial cell specific molecule-1 (esm1) (Supplementary Fig. 5b,c), genes previously shown to be enriched in angiogenic vessels[36,37]. In addition, we observed a significant upregulation of plgf, which encodes the Flt1-specific pro-angiogenic ligand Plgf, and lyve1, a gene expressed in veins and implied in lymphangiogenesis, in line with the venous expansion phenotype in $flt1^{ka601}$ mutants (Supplementary Fig. 5c)[38].

**Origin of endothelial cells in ectopic venous sprouts.** It is established that artery-derived ECs, on arteriovenous (AV) remodelling, contribute to the dorsal aspect of vISVs (ref. 39). Besides these remodelled artery-derived cells, another source may be PCV-derived venous endothelial cells as they can migrate over long distances[40]. However, a specific contribution of these venous ECs in populating the dorsal aspect of vISVs has not been shown thus far. To determine whether PCV-derived venous cells can colonize the dorsal aspect of vISVs, we performed cell tracking experiments using the $Tg(kdrl:nlskikGR)^{hsc7}$ transgenic line (Fig. 4a–i). A small part of the PCV was photo-converted at 30 hpf and individual venous endothelial cells were tracked in the period 30–60 hpf by time-lapse imaging (Fig. 4a–f, Supplementary Movie 5). We observed three scenarios (Fig. 4i). In scenario (I): PCV-derived venous endothelial cells migrated into the vISV and reached the most dorsal aspect of the vISV (Fig. 4c–e). In the dorsal aspect

of vISVs, PCV-derived endothelial cells were observed together with the remodelled artery-derived endothelial cells (Fig. 4f,g; artery-derived cells in green). Scenario (I), which we refer to as 'mixed' (both artery and vein-derived EC), accounted for 43.2% of cases (Fig. 4h,i). Of the mixed population 67.9% of endothelial cells were of venous origin and 32.1% of arterial origin (Fig. 4h, right panel). In scenario (II), the dorsal part of vISV only contained PCV-derived venous endothelial cells; artery-derived endothelial cells were absent. Scenario (II) accounted for 48.6% of cases (Fig. 4h,i). In scenario (III) we find that the dorsal part of vISV only contained artery-derived ECs; in this scenario the dorsal aspect of vISVs was not colonized by migrating PCV-derived venous endothelial cells (Fig. 4h,i). This scenario was observed in 8.2% of cases.

The $flt1^{enh}$ promoter marks ISV-ECs of arterial origin[39]. Loss of flt1 in Tg($flt1^{enh}$:Tdtomato; flt4:mCitrine) showed ectopic venous sprouts containing $flt1^{enh}$-expressing ECs (Fig. 4k,l,n–p). In the same embryo, we furthermore noted ectopic venous sprouts devoid of $flt1^{enh}$ expressing ECs (Fig. 4j,m,p), suggesting that these sprouts were only made of vein-derived ECs (Fig. 4m). To confirm a contribution of PCV-derived venous endothelium we performed cell tracking experiments in $Tg(kdrl:nlskikGR)^{hsc7}$ on loss of flt1 and indeed we found that PCV-derived venous ECs were capable of contributing to ectopic sprouting (Supplementary Fig. 6a). Interestingly, besides sprouts exclusively containing artery-derived, or venous-derived endothelium (Fig. 4m,o), we observed composite sprouts with artery and venous-derived endothelial cells juxtaposed (Fig. 4k,n).

**Vegfaa gain-of-function promotes venous sprouting.** Before 48 hpf trunk arterial sprouting is driven by Vegfaa and venous sprouting by Vegfc (refs 24,26,27). Since loss of flt1 mimics vegfaa gain-of-function, we expected changes in arterial branching in $flt1^{ka601}$. Rather surprisingly, we observed ectopic venous sprouting after 2.5 dpf (Fig. 3a,b; Fig. 5a,b,e). Primary artery development was not affected in flt1 mutants (Supplementary Fig. 6b,c,f,g), although primary arterial sprouts developed in close proximity to the neural tube (Supplementary Fig.6h–k).

Ectopic venous sprouting was conserved in several other vegfaa gain-of-function scenarios, including $vhl^{hu2114}$ mutants and $ptena^{-/-};ptenb^{-/-}$ double mutants (Fig. 5c–e). Von Hippel-Lindau protein (pVHL) is essential for the proteolytic degradation of Hif-1α, an evolutionary conserved transcription factor important for regulating vegfaa transcription[41,42]. Loss of vhl prevents Hif-1α degradation and augments vegfaa expression[41,42]. Accordingly, $vhl^{hu2114}$ mutants developed ectopic sprouts emanating from vISVs but not from aISVs (Fig. 5c,e). Changes in primary aISV sprouting were not observed (Supplementary Fig. 6d,f,g).

PTEN is a tumour suppressor gene acting as a PI3K/Akt signalling attenuator and linked to the progression of many tumours involving VEGF-A (refs 43,44). In zebrafish, two orthologues of pten exist, and $ptena^{-/-};ptenb^{-/-}$ double mutant zebrafish show increased vegfaa levels[44]. Detailed analysis of $ptena^{-/-};ptenb^{-/-}$ double mutants identified pronounced ectopic venous sprouting at the level of the neural tube (Fig. 5d,e). In pten double mutants ectopic venous sprout numbers were higher when compared with $flt1^{ka601}$ single mutant or $vhl^{hu2114}$ single mutant (Fig. 5e). In addition, in a small percentage of ISVs, $ptena^{-/-};ptenb^{-/-}$ double mutants displayed very few ectopic arterial sprouts (Fig. 5e).

Mechanistically, loss of vhl and flt1 augments Vegfaa function at different levels, through increased vegfaa transcription and higher Vegfaa protein bioavailability, respectively. We reasoned that combining both mutants should increase Vegfaa

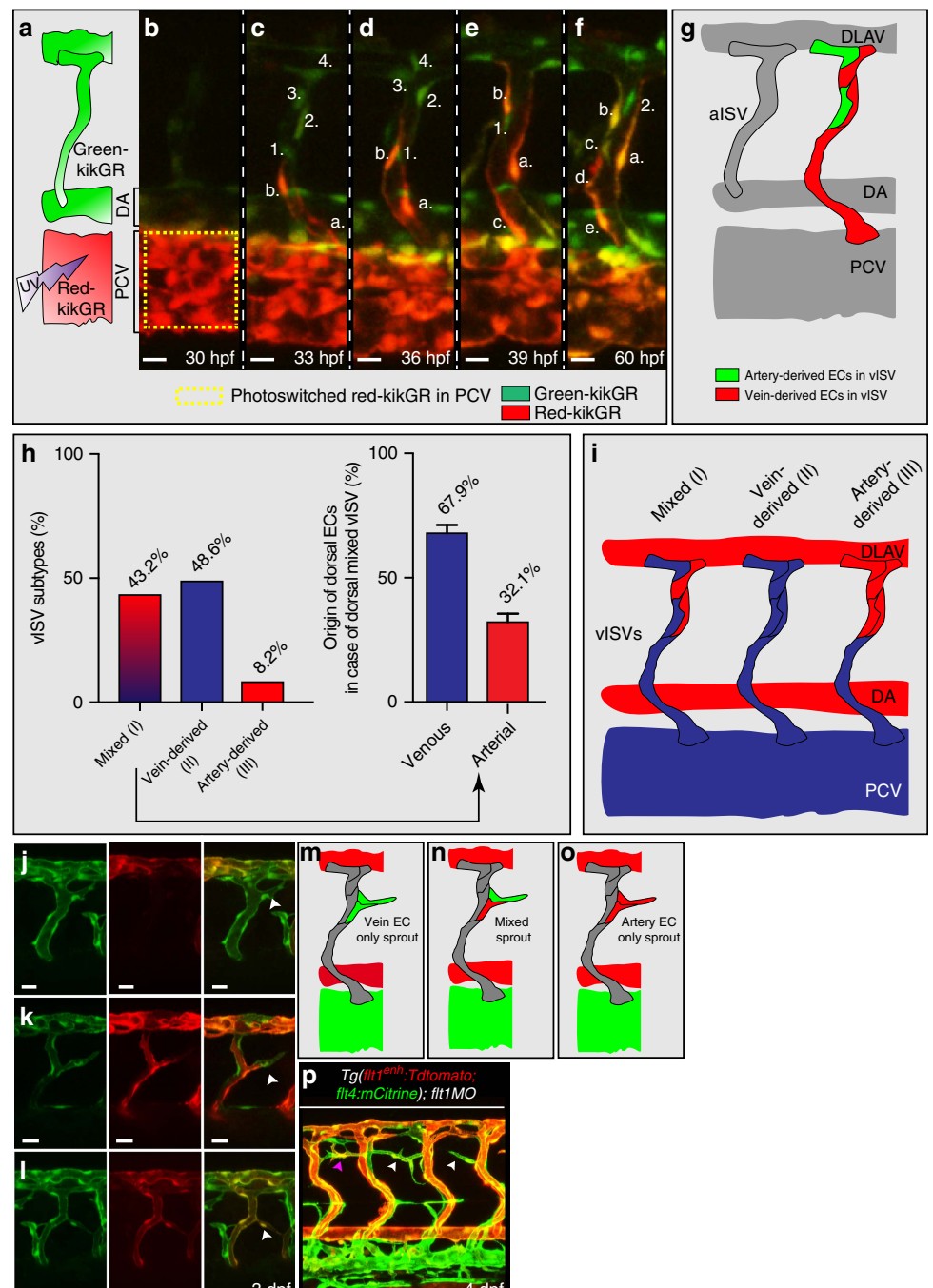

**Figure 4 | Arterial and venous-derived endothelial cells populate the dorsal part of venous ISV and contribute to sprouting upon loss of *flt1*.**
(**a**) Endothelial cell tracking in *Tg(kdrl:nlsKikGR)^hsc7* embryos. Photo-converted PCV-derived venous endothelial cells express red-kikGR. (**b–f**) Endothelial cell tracing during 30–60 hpf, showed that PCV-derived endothelium, indicated in red & labelled a,b,c,d,e, migrated along the ISV, from ventral to dorsal up to the most dorsal part of vISVs (cell labelled b). Pre-existing arterial endothelial cells in ISV, in green and labelled 1,2,3,4, shows artery-derived ECs in the dorsal part of vISVs (cell labelled 2) adjacent of PCV-derived EC (cell labelled **b,a**). (**g**) Schematic representation of the scenario imaged in **b–f**. (**h**) Left panel: Identity analysis of endothelial cells in the dorsal part of vISV revealed three different scenarios: (I) mixed, both arterial and venous-derived endothelium were present, (II) only vein-derived endothelium, (III) only artery-derived endothelium. (*n* = 10 experiments & 6ISVs/embryo). Right panel: % of artery and vein-derived endothelium in the mixed population scenario. (**i**) Schematic representation of the three identity scenarios in dorsal part of vISV. Arterial derived EC in red, venous-derived EC in blue. (**j–l**) Ectopic sprouting scenarios in *flt1* morphants in *Tg(flt1^enh:Tdtomato; flt4:mCitrine)*, *n* = 6 embryos. (**j**) Ectopic venous sprout devoid of *flt1^enh* expressing artery-derived EC (arrowhead). (**k**) Ectopic venous sprout (arrowhead) containing both arterial and venous-derived ECs; the *flt1^enh* expressing artery-derived EC is juxtaposed to the venous-derived cell at the tip (arrowhead). (**l**) Ectopic venous sprout only containing *flt1^enh* expressing artery-derived ECs; *flt1^enh* (red) and *flt4* (green) were expressed by the same cell which appears in yellow (arrowhead). (**m–o**) Schematic representation of the three ectopic venous sprouting scenarios. (**p**) Ectopic sprouting upon loss of *flt1* in *Tg(flt1^enh:Tdtomato; flt4:mCitrine)* (representative of 5 embryos). Flt1^enh positive sprouts (pink arrowhead) and sprouts devoid of *flt1^enh* (white arrowhead). MO, *flt1* morpholino, 1ng. aISV, arterial intersegmental vessel; DA, dorsal aorta; DLAV, dorsal longitudinal anastomotic vessel; PCV, posterior cardinal vein; vISV, venous intersegmental vessel. Scale bar, 20 μm in **j–l**, 10 μm in **b–f,p**.

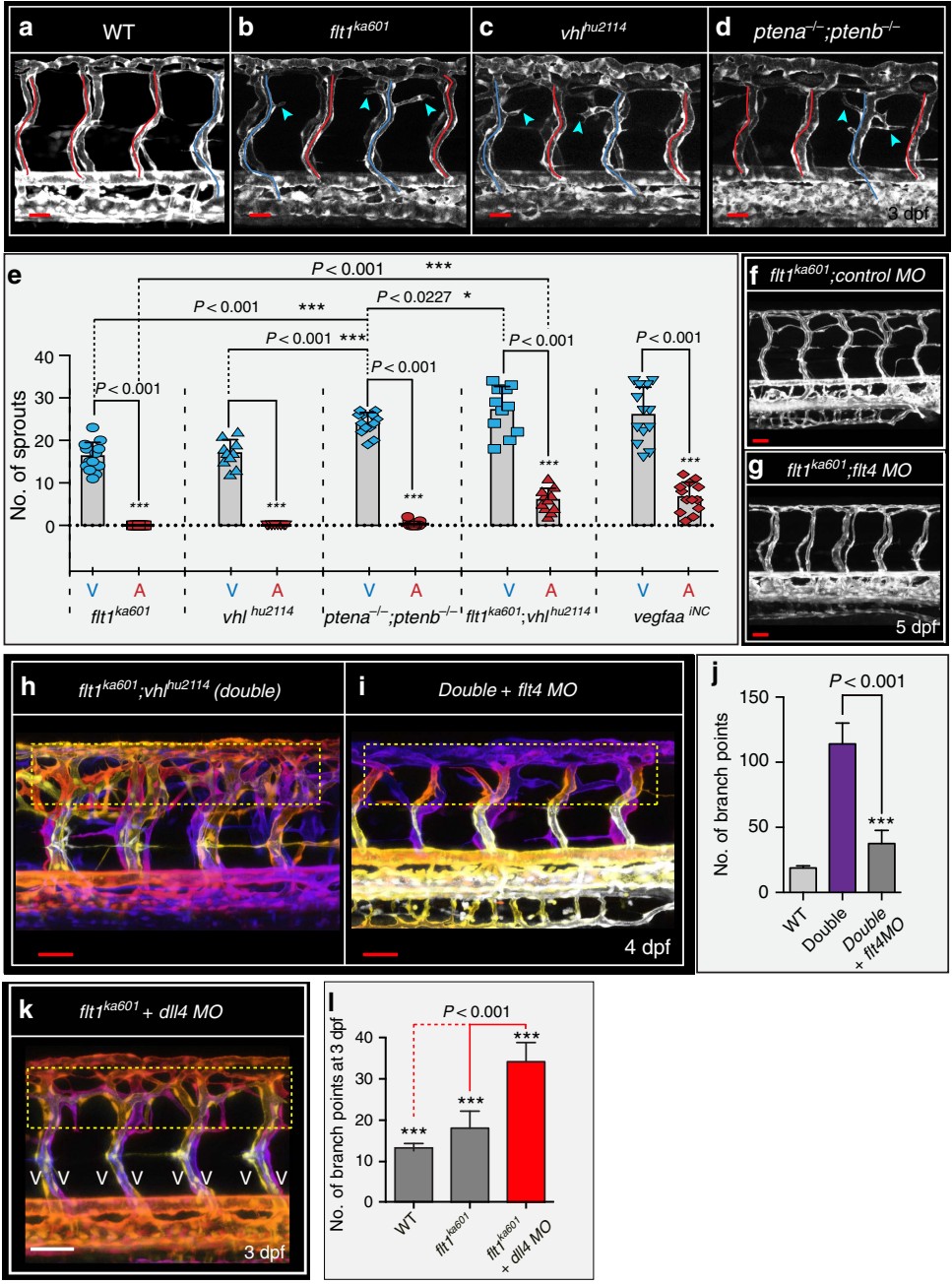

**Figure 5 | *flt1^{ka601}* mutants and *vegfaa* gain-of-function scenarios promote ectopic venous sprouting. (a–d).** Trunk vasculature at 3 dpf in WT (**a**), *flt1^{ka601}* (**b**), *vhl^{hu2114}* (**c**) and *ptena^{−/−};ptenb^{−/−}* double mutants (**d**) in *Tg(kdrl:EGFP)^{s843}* background. Note ectopic sprouts originate from vISVs (blue arrowheads) in mutants. aISVs indicated in red, vISVs in blue. (**e**) Quantification of ectopic sprouting in indicated mutants and inducible neuronal-specific *vegfaa* gain-of-function. In all models ectopic sprouting preferentially occurs in veins, mean ± s.e.m., n = 10–13/per group, t-test. (**f,g**) *flt1^{ka601}* mutants show hyper-branching and knockdown of *flt4* in *flt1^{ka601}* mutant rescues hyper-branching; n = 21 embryos per group. (**h,i**) Knockdown of *flt4* in *flt1^{ka601}; vhl^{hu2114}* double mutants (double) rescues hyper-branching; compare yellow dotted box in **h,i**. The position of vessels is colour-coded. Note: on loss of *flt4* the trunk vasculature consists almost exclusively of aISV. (**j**) Quantification of **h,i**. Mean ± s.e.m., n = 12 embryos per group, t-test. (**k,l**) Loss of *dll4* in *flt1^{ka601}* mutants augments ectopic branching compared with untreated *flt1^{ka601}* mutants. Note: on loss of *dll4* the trunk vasculature consists almost exclusively of vISV; n = 11 embryos per group, t-test. aISV, intersegmental artery; MO, morpholino; vISV, intersegmental vein. Scale bar, 30 μm in **a–d,f**; 50 μm in **f–i,k**.

and activate downstream Kdrl signalling even further. Indeed, *flt1^{ka601};vhl^{hu2114}* double mutants showed more severe hyper-branching of the trunk vasculature when compared with single mutants (Fig. 5e,h). Accordingly, *flt1^{ka601};vhl^{hu2114}* double mutants developed more ectopic venous sprouts when compared with either single *flt1^{ka601}* or single *vhl^{hu2114}* mutants (Fig. 5e). The *flt1^{ka601};vhl^{hu2114}* double mutants also developed a small number of ectopic arterial sprouts after 2.5 dpf (Fig. 5e).

However, venous sprout numbers were three times higher (P < 0.001) than arterial sprout numbers at this stage (Fig. 5e). Changes in primary aISV (24 hpf) sprouting were not observed (Supplementary Fig. 6e,f,g). Endoxifen-induced neuronal-specific overexpression of *vegfaa165* at 52 hpf in WT embryos also promoted ectopic venous sprouting (Fig. 5e; Supplementary Fig. 8e,f). In addition, a smaller number of ectopic arterial sprouts was noted, similar to *flt1^{ka601};vhl^{hu2114}*

double mutants (Fig. 5e). Taken together, ectopic venous sprouting was conserved in five *vegfaa* gain-of-function scenarios.

**Ectopic sprouting in *flt1^{ka601}* mutants requires veins.** To prove that in *flt1^{ka601}* mutant sprouts indeed emanated from veins, we interfered with early arterial-venous remodelling by blocking *flt4* (ref. 26) (Fig. 5f,g; Supplementary Fig. 6l). Loss of *flt4* in *flt1^{ka601}* mutants interfered with arterial-venous remodelling; as a consequence almost all trunk ISVs remained arterial[26] (Supplementary Fig. 6l). In line with the requirement for veins, the *flt1^{ka601}* hyper-branching phenotype was rescued (Fig. 5f,g). Furthermore, *flt4* loss-of-function in the *flt1^{ka601};vhl^{hu2114}* double mutants (denoted as double in Fig. 5h,i) also significantly reduced branching complexity (Fig. 5h-j; method quantification of branch points in Supplementary Fig. 1n). As ectopic sprouting requires venous endothelium, we next reasoned that promoting vISV formation in *flt1^{ka601}* mutants should augment branching. Vessel identity and Notch signalling are linked. In zebrafish, it is established that loss of the Notch ligand Dll4 promotes venous cell fate and *dll4* loss-of-function embryos display a trunk vasculature consisting almost exclusively of vISVs[45]. Accordingly, loss of *dll4* in *flt1^{ka601}* mutants significantly augmented ectopic branching when compared with control *flt1^{ka601}* mutants (Fig. 5k,l).

**Notch, pericytes and ectopic venous sprouting in *flt1^{ka601}*.** One explanation for the low arterial responsiveness in *vegfaa* gain-of-function scenarios may involve high arterial Notch activity since Notch acts as a repressor of sprouting in arteries, downstream of Vegfaa signalling[25,35,46]. To inhibit endothelial Notch signalling in arterial ISVs of *flt1^{ka601}* mutants, we expressed a dominant negative form of the Notch co-activator MAML (DN-MAML-EGFP) in an endoxifen inducible manner (Fig. 6a,b)[47]. We used the *flt1^{enh}* promoter construct which is mainly active in aISVs (ref. 39) to drive gal4-ERT2;UAS:DN-MAML-eGFP (notch^{iΔEC}) in *flt1^{ka601}* mutants. Transgene expression was initiated at 52 hpf by adding endoxifen. Endothelial-specific DN-*MAML* gain-of-function in *flt1^{ka601}* mutants induced ectopic aISV sprouting at the level of the neural tube (Fig. 6a,b,f). Even more pronounced ectopic arterial sprouting was observed with the γ-secretase inhibitor LY-411575 that blocks Notch activation; adding LY-411575 at 2 dpf activated ectopic arterial sprouting in *flt1^{ka601}* mutants (Fig. 6c-f). Venous sprout numbers were not significantly changed upon DN-MAML (16.1 ± 3.45 versus 17.1 ± 2.88) or LY-411575 treatment (15.9 ± 2.89 versus 14.2 ± 1.69). Addition of LY-411575 to WT at 2 dpf had no effect. To explain differential AV responsiveness, we also considered differences in pericyte cell coverage (Fig. 6g-j). Overall, pericytes were scarce with 88% of all ISVs investigated not being covered by pericytes. In the remaining 12% of cases, pericytes were found in both aISVs (9.94%) and vISVs (1.91%) along the ISV ventral-dorsal axis. In the most dorsal aspect of aISV and vISV, the region where ectopic sprouting occurs in *flt1^{ka601}*, pericytes were comparable between aISV and vISV (2.48% and 1.91% respectively, Fig. 6k).

**Vegf and Flt1 determine extent of spinal cord vascularization.** Neurons expressed *vegfaa* (Supplementary Fig. 1), and neuronal cells of both 3 dpf WT and *vhl* loss-of-function embryos had significantly higher *vegfaa* levels than non-neuronal cells (Fig. 7a,b; FACS settings in Supplementary Fig. 7a-d). Furthermore, neuronal *vegfaa* expression was significantly increased in *vhl* loss-of-function when compared with WT (Fig. 7a,b). Thus, at this stage of development neurons are

the major source of *vegfaa,* and not other tissues like developing muscle[48]. We next examined whether neurons can direct sprouts into the neural tube (Fig. 7c-i). We compared the *flt1^{ka601};vhl^{hu2114}* double mutant (Fig. 7c,d) with *flt1^{ka601}* mutant and WT and found striking changes in optical sections of the neurovascular interface (Fig. 7e-h). In *flt1^{ka601}*, sprouts occasionally projected into the neural tube (Fig. 7g), whereas in *flt1^{ka601};vhl^{hu2114}* double mutants many branches invaded the neural tube (Fig. 7h,i).

In the mutants with *vegfaa* gain-of-function, the spinal cord becomes vascularized relatively early, between 3 and 4 dpf. In WT, the spinal cord is vascularized much later in development starting in the period between 12 and 14 dpf (Fig. 7j-l). In those older WT embryos, sprouts preferentially emanated from venous ISVs, displayed nuclear positioning as described for the *flt1* mutant (Supplementary Fig. 7e,f) and the onset of vascularization of the WT spinal cord coincides with decreased *sflt1* expression during this stage of development (Supplementary Fig. 7g).

**Neuronal sFlt1 and Vegfaa regulate sprouting from veins.** We next generated tissue-specific and inducible *flt1* and *vegfaa* gain-of-function models. Loss of neuronal sFlt1 in *flt1^{ka601}* mutants may augment neuron-derived Vegfaa availability and promote ISV sprouting. Hence, restoring neuronal sFlt1 in *flt1^{ka601}* mutants should provide a rescue, whereas neuronal-specific *flt1* loss-of-function should induce hypersprouting. To test the first scenario we expressed -3.2elavl3:gal4-ERT2;UAS:GFP-p2a-sflt1 (sflt1^{iNC}) in *flt1^{ka601}* mutants (Fig. 8a-d, branch quantification method in Supplementary Fig. 1n). This construct allows precise time-controlled expression of *sflt1* specifically in neurons. We found that transgene activation in neurons at 52 hpf, just before the emergence of the ectopic sprouts in *flt1^{ka601}* mutants, rescued the vascular hyper-branching phenotype (Fig. 8b-d).

We next explored whether neuron-specific loss of *flt1* is sufficient to induce ISV hyper-branching (Fig. 8e-i). To accomplish neuron-specific loss of *flt1* we expressed the *flt1* targeting sgRNA^{flt1E3} (U6:sgRNA^{flt1E3}, the same sgRNA as used to generate *flt1^{ka601}* mutants; expressed in all cells) together with the Cas9 construct employing the Gal4-UAS system under the control of the pan-neuronal promoter *Xla.Tubb* (-3.8Xla.Tubb:gal4-VP16/UAS:Cas9-t2a-eGFP (flt1^{ΔNC}); (Fig. 8e)[49]. To optimize the biallelic knockout efficacy, we injected the construct into embryos heterozygous for the *flt1* − 1 nt allele (flt1^{ka601/+}). GFP signal was detected in spinal cord neurons indicating efficient *Xla.Tubb*-driven neuron-specific expression of Cas9 (Fig. 8h). Neuronal loss of *flt1* significantly induced ectopic venous sprout formation when compared with WT and *flt1^{ka601/+}* heterozygous mutants (Fig. 8f-i). In contrast, sprouting was not observed when Cas9 was expressed under a vascular promoter (Supplementary Fig. 8a) or in embryos only carrying the sgRNA without Cas9.

To substantiate the contribution of neuronal *sflt1* we next employed multiplexed custom designed miRNAs directed against *sflt1* 3′UTR arranged with a common miR-155 backbone[50] (Supplementary Fig. 8b). The constructs were expressed under control of vascular (*flt1^{enh}*) and neuronal (*Xla.Tubb*) specific promoters. Targeting neuronal *sflt1* resulted in ectopic sprouting (Supplementary Fig. 8c), but targeting vascular *sflt1* failed to induce sprouts (Supplementary Fig. 8d).

Next we performed cell transplantation experiments, which demonstrated that neuronal *flt1* and not vascular *flt1* is the physiologically relevant mediator of sprouting at the level of the neural tube (Fig. 8j-l). Transplantation of *flt1* mutant neurons into WT hosts induced ectopic sprouting (Fig. 8k).

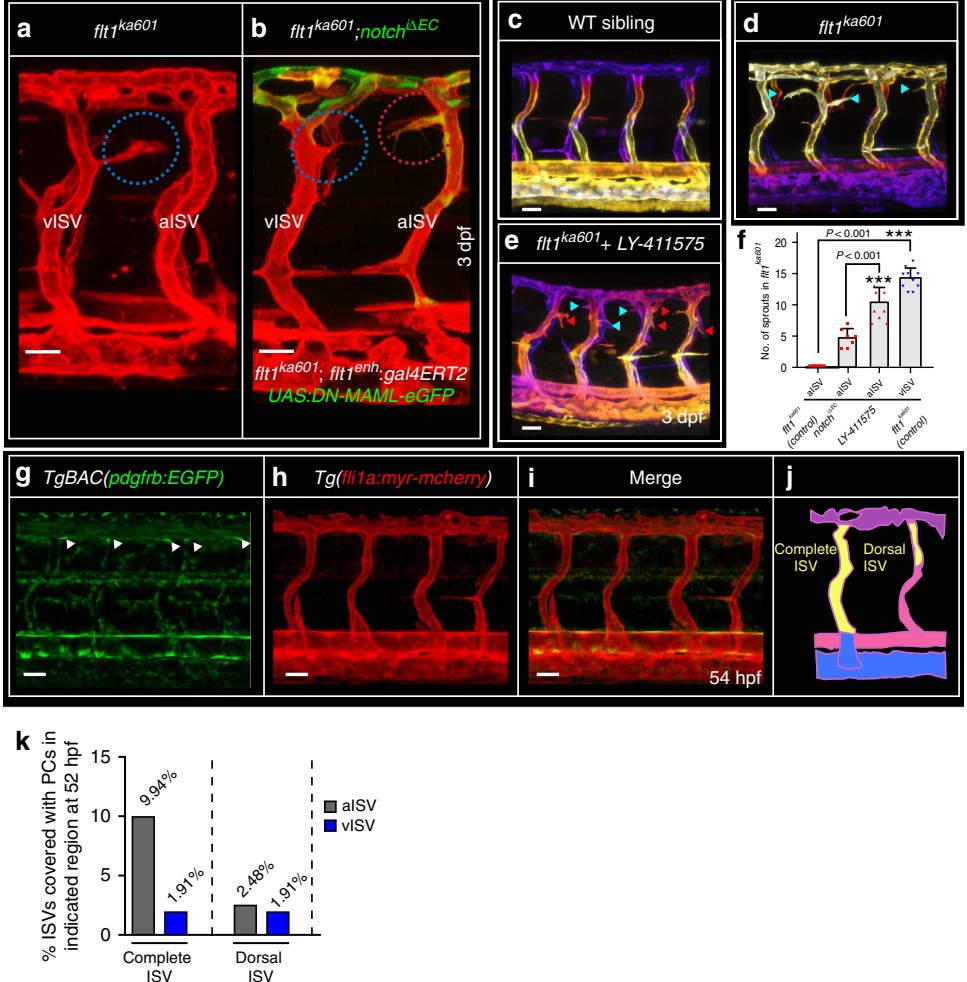

**Figure 6 | Notch inhibits ectopic arterial sprouting in *flt1^ka601* mutants.** (**a**), *flt1^ka601* mutants show ectopic venous sprouts (blue circles), but no arterial sprouts. (**b**) Inhibiting arterial Notch by endoxifen-induced arterial ISV-specific expression of dominant negative *MAML-eGFP* (Notch ^iΔEC) at 52 hpf under control of the *flt1^enh* promoter in *flt1^ka601* mutant results in the emergence of ectopic arterial sprouts (red circles); representative image from 7 experiments. (**c–e**) Trunk vasculature of WT (**c**), *flt1^ka601*(**d**) and *flt1^ka601* treated with Notch inhibitor LY-411575 (**e**). LY-411575 was added at 2 dpf. Note the emergence of ectopic arterial sprouts upon LY-411575 treatment. (red arrowhead: arterial sprout; blue arrowhead: venous sprout). (**f**) Quantification of experiments in (**a–e**), mean ± s.e.m., $n = 7$ for Notch^iΔEC, $n = 10$ for LY-41157 treatment, $n = 10$ for *flt1^ka601*; t-test. (**g–i**) Imaging of pericytes in *TgBAC(pdgfrb:EGFP);Tg(fli1a:myr-mcherry)* double transgenic at 54 hpf. (**j**) Schematic representation of pericyte number counting in ISVs as performed in (**k**). (**k**) Quantification of pericyte recruitment in aISVs and vISVs at 54 hpf. ($n = 246$ ISVs from 14 embryos). ISV, intersegmental artery; Notch ^iΔEC, inducible ISV-specific loss of Notch; vISV, intersegmental veina. Scale bar, 25 μm in **a–e,g–i**.

In contrast, transplantation of *flt1* mutant endothelial cells into WT hosts failed to induce sprouting (Fig. 8l).

To prove that neuron-derived Vegfaa promotes hyper-branching, we generated neuronal tissue-specific and inducible *vegfaa165* gain-of-function zebrafish (Supplementary Fig. 8e,f; quantification in Fig. 5e). Transgenic expression was initiated by adding endoxifen after completion of AV remodelling at 52 hpf. In this scenario hyper-branched neovascular networks formed at the level of the neural tube, similar to *flt1^ka601* mutants (Supplementary Fig. 8e,f). Neuronal *vegfaa121* was also capable of inducing sprouting (Supplementary Fig. 8g). In contrast, neuron-specific and inducible *vegfc* gain-of-function, induced at 54 hpf, did not induce ectopic sprouts (Supplementary Fig. 8h). Timing of transgene expression was relevant as inducible neuron-specific *vegfaa165* overexpression prior to completion of AV remodelling resulted in thickened abnormal vascular structures (Supplementary Fig. 8i,j). In the same line, neuron-specific constitutive overexpression of *sflt1* completely annihilated ISV formation (Supplementary Fig. 8k).

To confirm that the *flt1^ka601* phenotype involved gain of Vegfaa, we titrated *vegfaa* levels using a low dose *vegfaa* targeting morpholino[51]. Reducing *vegfaa* in *flt1^ka601* mutants rescued the hyper-branching phenotype (Fig. 8m–o). Vegfaa signals via Kdrl and application of ki8751, an established Kdrl tyrosine kinase inhibitor in zebrafish[52] to *flt1^ka601* mutants at 2.5 dpf annihilated the formation of the ectopic neovascular networks (Supplementary Fig. 8l,m,o). In contrast, the Flt4-specific tyrosine kinase inhibitor MAZ51 (ref. 52) did not rescue hyper-branching in *flt1^ka601* (Supplementary Fig. 8l,n,o). Vegfaa-driven primary artery sprouting can occur in the absence of blood flow perfusion. To test if Vegfaa-driven ectopic venous sprouting in *flt1^ka601* mutants is affected by blood flow, we modulated cardiac activity and flow with BDM or tricaine. We found that loss of flow completely rescued ectopic hyper-sprouting in *flt1^ka601* mutants (Supplementary Fig. 9a-e). Inhibition of PI3 kinase with wortmannin also significantly reduced ectopic sprouting (Supplementary Fig. 9f).

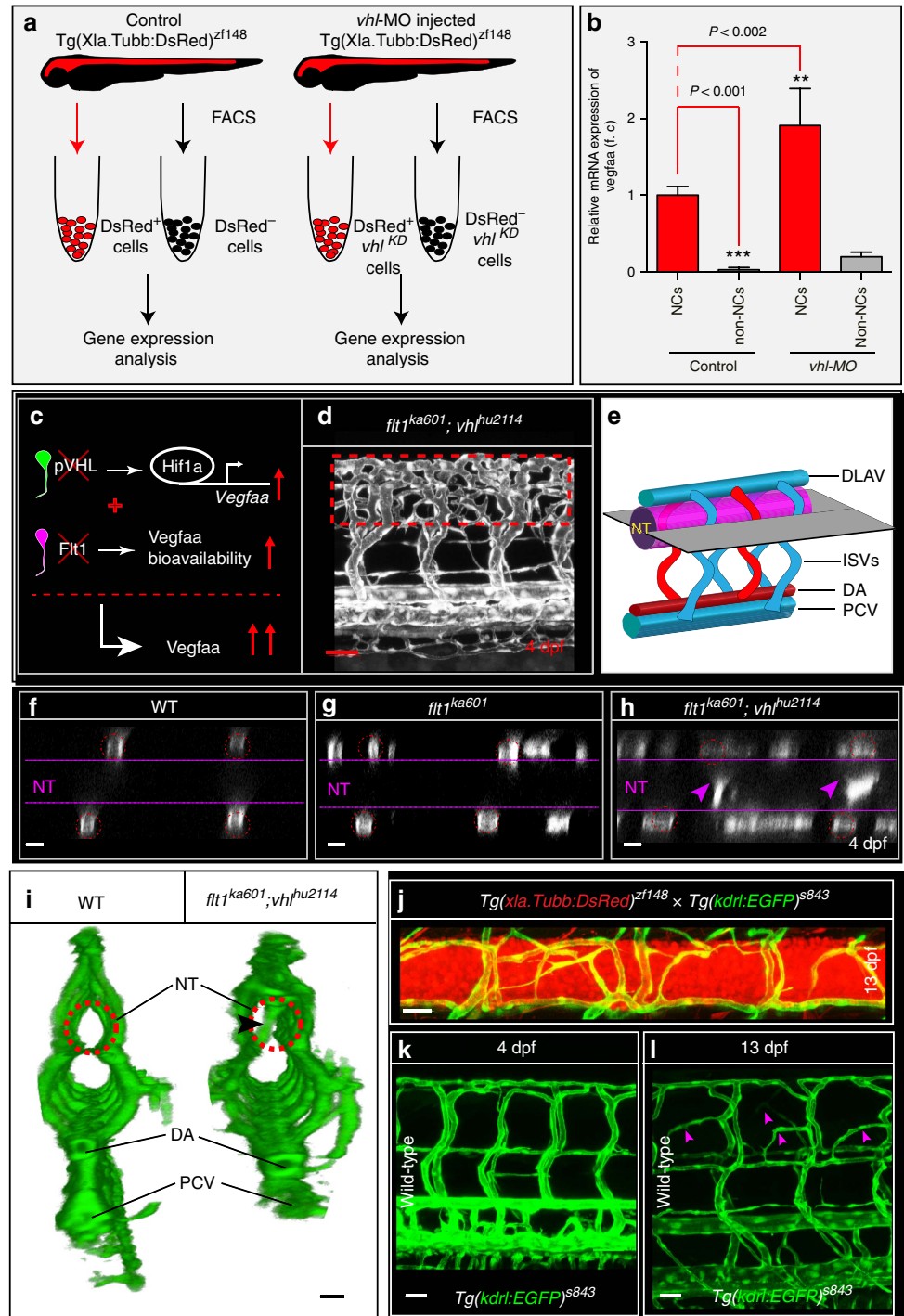

**Figure 7 | Neurons are a major source of Vegfaa and attract sprouting vessels.** (**a**) FACS procedure for obtaining neuronal cells in control and *vhl* morphants using *Tg(Xla.Tubb:DsRed)^zf148^* neuronal reporter embryos. (**b**) Quantification of *vegfaa* expression using real-time qPCR in FAC-sorted cell populations at 3 dpf. Note that neuronal cells expressed significantly more *vegfaa* than non-neuronal cells. Loss of *vhl* promoted neuronal *vegfaa* expression. Mean ± s.e.m., n = 3 separate experiments in triplicate (two-way ANOVA). (**c**) Schematic representation: loss of *vhl* augments *vegfaa* transcription, loss of *flt1* augments Vegfaa bioavailability; combining both mutants augments Vegfaa bioavailability above single mutant level. (**d**) Trunk vasculature in *flt1^ka601^;vhl^hu2114^* double mutants at 4 dpf. Note the severe hyper-branching at the level of the neural tube, red-dotted box. (**e**) Schematic representation of optical section (shown in **f**–**h**) through the neural tube and associated trunk vasculature. (**f**–**h**) Dorsal view on optical section through WT (**f**), *flt1^ka601^* (**g**) and *flt1^ka601^;vhl^hu2114^* double mutants (**h**). Note invasion of sprouts into the neural tube in double mutants (arrowheads in **h**). Red circle indicates position of ISVs, dotted line neural tube boundary. (**i**) Transverse 3D-rendered view of vasculature (green) through the trunk in WT (left panel) and *flt1^ka601^;vhl^hu2114^* double mutants (right panel); note vessels penetrating the neural tube in mutant (compare vessel in dotted circle right panel, arrowhead; such vessels are absent in WT left panel; representative image from 3 separate experiments). (**j**) Representative image of spinal cord vascular network in *Tg(xla.Tubb:DsRed)^zf148^; Tg(kdrl:EGFP)^s843^* double transgenic at 13 dpf. (**k**,**l**) Comparison of trunk vasculature in WT at 4 dpf (**k**) and at 13 dpf (**l**); note the emergence of ectopic branches (pink arrowheads) at level of the spinal cord. DA, dorsal aorta; f.c. fold change; KD, knockdown; NT, neural tube; NC, neuronal cell; PCV, posterior cardinal vein. Mutants are in *Tg(kdrl:EGFP)^s843^* background. Scale bar, 50 μm in **d**; 25 μm in **f**–**l**.

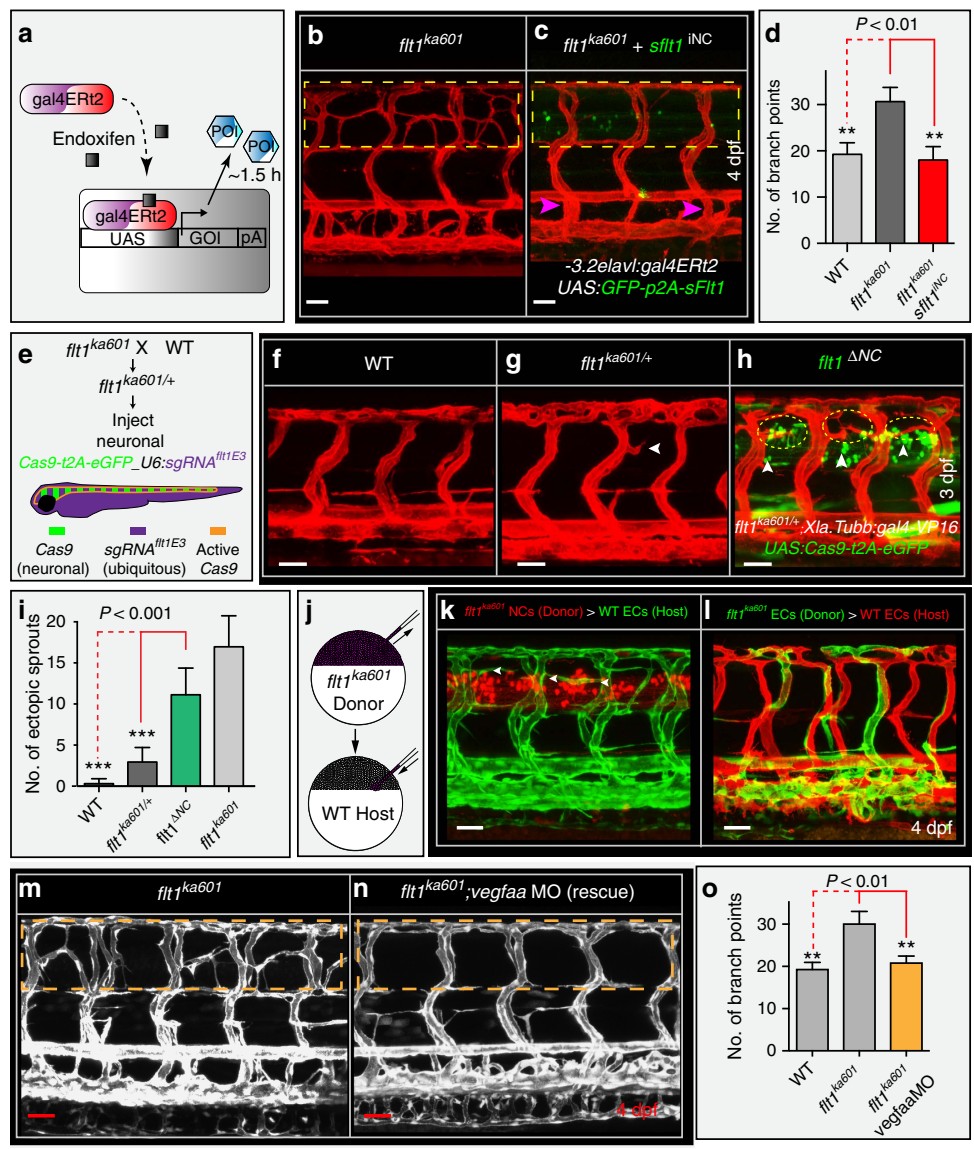

**Figure 8 | Neuronal Flt1 regulates vascular branching by titrating neuronal Vegfaa.** (**a**) Schematic representation of endoxifen inducible gain-of-function approach in zebrafish. In the present situation Gal4 is under the control of neuron-specific promoters *elavl* or *Xla.Tubb*. Expression can be observed within 1.5 h upon endoxifen application. (**b**) Hyper-branching in *flt1^{ka601}* mutants (dotted box). (**c**) Endoxifen inducible neuron-specific *sflt1* gain-of-function rescues hyper-branching in *flt1^{ka601}* mutants; compare dotted box in c and b. Purple arrowheads indicate vISVs; endoxifen was applied at 52 hpf. (**d**) Quantification of rescue in (b,c), mean ± s.e.m, n = 15–19 embryos per group. (**e**) Approach for generating a neuron-specific *flt1* mutant. Cas9 was expressed under control of neuronal promoter *Xla.Tubb*; sgRNA was expressed ubiquitously, resulting in Cas9 activity in neuronal cells only (domain marked by orange border). Heterzygous *flt1^{ka601/+}* were used to facilitate biallelic knockout. (**f–h**) Neuron-specific loss of *flt1* (*flt1^{ΔNC}*) induces ectopic sprouting (**h**), sprouts in yellow dotted ellipse, arrowheads indicate neuronal cells with Cas9 expression. (**i**) Quantification of ectopic sprouting for indicated genotypes. Note that neuron-specific loss of *flt1* significantly augments ectopic sprouting (green bar) mean ± s.e.m, n = 16 embryos per group, t-test. (**j–l**) Transplantation of *flt1* mutant neuronal cells (**k**) and endothelial cells (**l**) into WT. Note: transplantation of *flt1* mutant neuronal cells induced sprouting (k, arrowheads); 9 out of 12 neuronal cell transplantations resulted in sprout formation. In all 10 endothelial cell transplantations, sprouts were absent (**l**). (**m,n**) Low dose morpholino-mediated reduction of *vegfaa* expression in *flt1^{ka601}* mutants rescues sprouting defects; compare dotted box in (**m,n**). (**o**) Quantification of rescue in (m,n), mean ± s.e.m., n > 5 per group, t-test. DA, dorsal aorta; PCV, posterior cardinal vein; DLAV, dorsal longitudinal anastomotic vessel; NT, neural tube. GOI, gene of interest; POI, protein of interest; iNC, inducible, neuronal cell specific gain-of-function; ΔNC, neuron-specific loss of *flt1*; MO, morpholino. Scale bar, 50 μm in **b–h,m,n**; 25 μm in **k,l**.

## Discussion

Intimate cross-talk between vessels and the nervous system is important for tissue homeostasis. During embryonic development, neuronal stem cells differentiate into mature neurons, a process that associates with a change in cellular metabolism[53]. Concomitantly with developmental neurogenesis, changes occur in the vascular network feeding the spinal cord. We show in the zebrafish embryo that neurons in the developing spinal cord express the pro-angiogenic ligand Vegfaa and anti-angiogenic soluble Vegf receptor-1, sFlt1, which acts as a Vegfaa scavenger (Fig. 9). Spinal cord neurons are in close contact to the developing trunk vasculature, and we show that these vessels are responsive to changes in neuronal sFlt1 and Vegfaa. Using a combination of global and tissue-specific

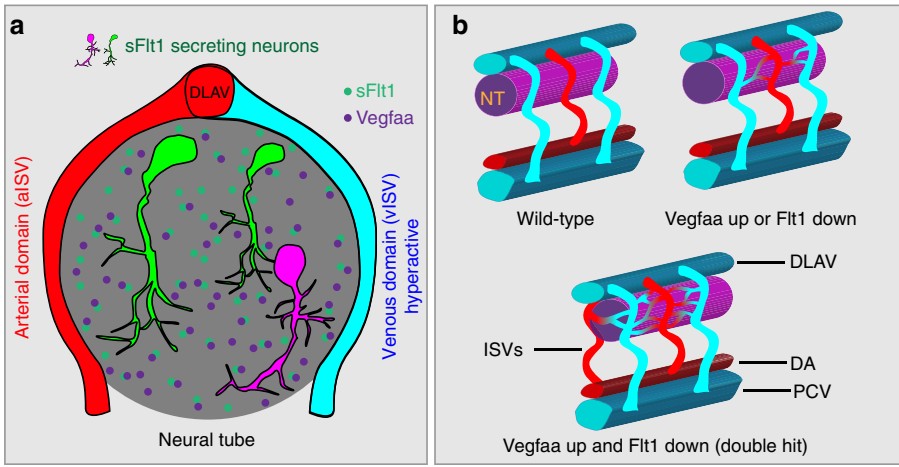

**Figure 9 | Schematic representation of neurovascular communication involving neuronal sFlt1-Vegfaa and sprouting from intersegmental veins.**
(**a**) Spinal cord neurons produce both sFlt1 and Vegfaa in close proximity to the dorsal aspect of intersegmental arteries and veins. (**b**) Schematic representation of vascularization around the neural tube in WT (top left), *flt1*[ka601] single mutant or *vhl*[hu2114] single mutant (top right), and *flt1*[ka601];*vhl*[hu2114] double mutant (bottom). Loss of *flt1* or *vhl* induces the formation of a peri-neural tube network, and combining both mutants in addition promotes sprouting into the neural tube. NT, neural tube; ISV, intersegmental vessel (a-arterial, v-venous); DA, dorsal aorta; PCV, posterior cardinal vein; DLAV, dorsal longitudinal anastomotic vessel; hpf, hours post fertilization.

loss of *flt1* mutants, and further substantiated by *vegfaa* loss- and gain-of-function experiments as well as cell transplantations, we demonstrate that neuronal sFlt1 restricts neuronal Vegfaa and vessel branching morphogenesis at the neurovascular interface. Differential regulation of *vegfaa* and *sflt1* allows orchestration of the onset and extent of spinal cord vascularization (Fig. 9). We propose that neurons may use sFlt1-Vegfaa to adjust vascularization according to their developmental needs.

*Flt1*[ka601] mutants develop ectopic sprouts emanating from venous ISVs around embryonic day 2.5. Neuron-specific targeting of *flt1* or *sflt1* using CRISPR/Cas9- and miRNA-based approaches respectively, result in *flt1*[ka601] comparable phenotypes. Transplantation of *flt1* mutant neurons into WT hosts induces ectopic sprouting which is not observed after transplantation of *flt1* mutant endothelial cells, suggesting that neuronal *flt1* is the physiologically relevant mediator in our mutant. Neuronal-specific gain of *sflt1*, reducing *vegfaa* levels, or inhibition of Kdrl signalling provide a rescue suggesting that ectopic venous sprouting is mediated by the Vegfaa-Kdrl signalling axis. Accordingly, ectopic venous sprouting from the dorsal aspect of vISVs is conserved in five independent *vegfaa* gain-of-function scenarios.

Previous studies have shown that during AV remodelling, aISV-derived endothelial cells remain integrated in the dorsal aspect of vISVs (ref. 39). We confirm that remodelled artery-derived endothelial cells indeed contribute to this domain although they are not the sole or most important endothelial source. Using *in vivo* cell tracking we find posterior cardinal vein-derived endothelial cells migrating against the direction of blood flow to populate venous ISVs including the dorsal aspect where sprouts are formed. Here, venous-derived endothelium can co-exist with the artery-derived endothelium. With respect to the endothelial cells populating the dorsal part of vISVs, our data now reveal three different scenarios. The dorsal aspect can contain a mix of both artery and venous-derived endothelium (43%), only vein-derived endothelium (48%) or only artery-derived endothelium (8%). On loss of *flt1*, both artery- and vein-derived endothelium give rise to ectopic vISV sprouts. This prompts toward the concept that integration into the local venous ISV environment constitutes a permissive factor for sprouting, regardless of the endothelial origin.

*Flt1*[ka601] mutants display ectopic sprouting in vISVs but not in aISVs, indicating that AV vessel identity or compartment-specific cues may be involved in the novel sprouting type described here. Notch is tightly linked to both AV vessel specification and sprouting, as Notch programs arterial identity and Notch signalling represses sprouting of arteries[25,45]. Lack of Notch associates with venous identity and Notch signalling is absent in venous ISVs (ref. 35). Interference with AV remodelling or Notch indeed affects sprouting numbers in *flt1*[ka601]. Inhibiting the remodelling of arterial ISVs into venous ISVs in *flt1*[ka601], and thus creating a trunk vasculature that almost exclusively consists of aISVs, rescues ectopic hypersprouting. Conversely, promoting venous cell fate and creating a trunk that consists of vISVs by knock-down of *dll4* augments vascular branching in *flt1*[ka601]. Inhibiting Notch by endothelial expression of *DN-MAML* or pharmacological treatment with LY-411575 stimulate ectopic arterial sprouting in *flt1*[ka601], without affecting venous sprout numbers. Thus active Notch signalling in arteries most likely accounts for the observed AV sprouting differences. We propose that the artery-derived endothelial cells that become incorporated in vISVs on AV remodelling are relieved from Notch and adopt a venous fate, enabling them to respond to changes in local Vegfaa.

Our data show that vISVs contribute to the vascularization of the spinal cord with Vegfaa-Kdrl signalling mediating vISV branching events. Interestingly, in other domains of the zebrafish trunk venous vasculature, endothelial cells are responsive to alternative signals. The Vegfc-Flt4 signalling pathway drives secondary venous sprouting from the PCV contributing to the formation of the lymphatic vasculature. BMP-Smad signalling has been shown to promote venous sprouting in the caudal vein plexus[54]. Furthermore, PCV-derived endothelial cells contribute to the formation of the gut vasculature[55]. Regeneration of the tail fin vasculature upon injury also starts from the venous side[56]. The concept emerging from these observations is that initiation of organ vascularization is initiated or proceeds from the venous vasculature. The mouse embryo coronary vasculature is vein-derived, and coronary arteries form by developmental reprogramming of venous endothelium[57]. Context-dependent heterogeneity in signalling mechanisms establishing venous branching, may allow versatile control of tissue vascularization in a spatio-temporal manner.

Using sFlt1 as a rheostat to control Vegfaa bioavailability, constitutes a means to regulate Vegfaa independent of *vegfaa* promoter activity, *vegfaa* mRNA or protein stability. We propose that this enables neurons to dynamically fine-tune the extent and onset of peri-neural vascular network formation and sprouting into the spinal cord. While the peri-neural network may serve to sustain growth of the developing nervous system, vessel sprouting into the spinal cord and relief of hypoxia has been associated with changes in neuronal stem cell metabolism, triggering differentiation events[53]. Therefore, untimely or excessive vascularization of the spinal cord is potentially harmful as it may promote premature stem cell differentiation and disrupt the carefully orchestrated neuronal specification process. We propose a two-tiered checkpoint mechanism involving sFlt1 and Vegfaa, requiring two decisions to guide vascularization, namely Vegfaa up- and sFlt1 downregulation, to protect neurons from harmful angiogenesis and oxygen variations during early stages on the one hand, and on the other hand to enable more mature neurons to attract sufficient vessels into the spinal cord after stem cell differentiation has been completed.

## Methods

### Ethics statement.
Zebrafish husbandry and experimental procedures were performed in accordance with the German animal protection standards and were approved by the government of Baden-Württemberg, Regierungspräsidium Karlsruhe, Germany (Akz.: 35-9185.81/G-93/15).

### Transgenic lines.
$Tg(fli1a:EGFP)^{y1}$, $Tg(kdrl:hsa.-HRAS-mcherry)^{s916}$, TgBAC $(flt1:YFP)^{hu4624}$, $Tg(fli1a:nGFP)^{y7}$, $Tg(Xla.Tubb:DsRed)^{zf148}$, $Tg(kdrl:EGFP)^{s843}$, $Tg(HuC:EGFP)^{as8}$, $Tg(mnx1:GFP)^{ml2}$, $Tg(flt1^{enh}:tdTomato)^{hu5333}$, $Tg(flt4: mCitrine)^{hu7135}$, $Tg(kdrl:nlskikGR)^{hsc7}$, $Tg(fli1a:myr-mcherry)$, TgBAC(pdgfrb:EGFP) as well as $vhl^{hu2114}$ and $ptena^{-/-};ptenb^{-/+}$ mutants were used as published[39,44,58–61].

### Morpholino injections.
Morpholino antisense oligomers (MOs; Gene Tools) were prepared at a stock concentration of 1 mM according to the manufacturer. MOs were injected into the yolk of one-cell stage embryos. We used the flt4 ATG MO, 5′-CTCTTCATTTCCAGGTTTCAAGTCC-3′ (4 ng), the flt1 ATG MO, 5′-ATATCGAACATTCTCTTGGTCTTGC-3′ (1 ng or 3 ng), the vhl e1i1 splice MO 5′-GCATAATTTCACGAACCCACAAAAG-3′ (6 ng), the vegfaa ATG MO 5′-GTATCAAATAAACAACCAAGTTCAT-3′ (0.3 ng), the dll4 MO 5′-TAGGGTTTAGTCTTACCTTGGTCAC-3′ (6 ng), and a control MO 5′-CTCTTACCTCAGTTACAATTTATA-3′ (10 ng) (refs 18,26,45,51,62).

### mRNA injection and generation of transgenic and mutant lines.
For the generation of mutants 1 nl of a mixture containing 600 ng μl$^{-1}$ capped and polyadenylated Cas9-nls mRNA and 50 ng μl$^{-1}$ sgRNA was injected into one-cell stage embryos[63]. Cas9 mRNA was produced by in vitro transcription from the MLM3613 plasmid using the mMessage mMachine T7 Ultra Kit (Ambion). The MLM3613 plasmid was a gift from Keith Joung (Addgene plasmid #42251). For the generation of transgenic lines 1 nl of a mixture of 12.5 ng μl$^{-1}$ transposase mRNA and 25 ng μl$^{-1}$ plasmid DNA was injected into one-cell stage embryos.

### Cell transplantations.
Cell transplantations were performed using 3.5 hpf donor and host blastula-stage embryos. Approximately 50–100 cells were taken from the donor's animal pole and transferred close to the host's lateral marginal zone (for ECs) or slightly above for neuronal cells. Donors and hosts carried distinct neuronal and endothelial-specific reporters to identify the source of ECs and neurons within chimeras.

### Generating flt1 mutants.
The zebrafish flt1 gene consists of 34 exons encoding membrane-bound flt1 (mflt1) and a shorter soluble flt1 (sflt1) form. Soluble flt1 is generated through alternative splicing of flt1 mRNA at the exon 10 - Intron 10 boundary (Supplementary Fig. 2a). To annihilate the production of both mflt1 and sflt1 and obtain flt1 mutants, we targeted exon 3, using a CRISPR/Cas approach. We designed five sgRNAs targeting exon 3, encoding the extracellular Ig1 domain relevant for Vegfaa binding. Oligonucleotides containing the GG-N18 targeting sequence and overhangs were purchased from Eurofins (Ebersberg, Germany). The annealed oligos were ligated into DR274 which was a gift from Keith Joung (Addgene plasmid # 42250)[63]. The corresponding genomic region (surrounding exon 3) was amplified by PCR using primer pair Flt1_E3_gDNA_r and Flt1_E3_gDNA_f and indels were quantified with

T7EI assay or direct Sanger sequencing of the PCR product as described (for primer sequences see Supplementary Table 6)[63]. The T7EI cleavage products of 211 and 249 bp were quantified using ImageJ. The sgRNA$^{flt1E3}$ (Supplementary Table 4) with the highest cleavage rate (~70%) was used to generate the flt1 mutants. WT embryos were coinjected with sgRNA$^{flt1E3}$ plus capped and polyadenylated Cas9 mRNA. Four independent lines with frame shift mutations were investigated in more detail. The flt1$^{ka601}$ (exon 3 -1 nt allele), flt1$^{ka602}$(exon 3 -5 nt allele), flt1$^{ka603}$(exon 3 + 5 nt allele) and flt1$^{ka604}$(exon 3 -14 nt allele) have a premature termination codon (PTC) resulting in a truncated protein devoid of a functional extracellular Vegfaa binding domain. Embryos carrying the mutation were raised and outcrossed to vascular and neuronal reporter lines ($Tg(kdrl:eGFP)^{s843}$, $Tg(fli1a:eGFP)^{y1}$, $Tg(fli1a:nGFP)^{y7}$, $Tg(kdrl:hsa.HRAS-mcherry)^{s916}$, and $Tg(Xla.Tubb:DsRed)^{zf148}$).

### Generation of mflt1-specific mutants.
To generate mflt1 mutants we used a CRISPR/Cas approach and designed an sgRNA targeting E11b, the first specific mflt1 exon[18]. In this scenario splicing of intron 10 and exon 11a relevant for generating sflt1 mRNA remains unaffected. Oligos Flt1E11_O1_A_15 and Flt1E11_O2_A_15 were annealed and cloned into DR274 as described for flt1 mutants. Founders were identified by PCR and subsequent Sanger sequencing, using primers Flt1E11A2386576F and Flt1E11A2386151R. We identified four frame shift mutants harbouring a PTC in exon 11b. Flt1$^{ka605}$(exon 11b + 28 nt), flt1$^{ka606}$(exon 11b + 20 nt), flt1$^{ka607}$(exon 11b − 1 nt) and flt1$^{ka608}$ (exon 11b − 1 nt and one mutation) mflt1 mutants were outcrossed to $Tg(kdrl:EGFP)^{s843}$ and $Tg(Xla.Tubb:DsRed)^{zf148}$. All four mflt1 mutants were phenotypically comparable and in this manuscript only the mflt1 mutant flt1$^{ka605}$ is shown. All sgRNA sequences and oligos used for annealed oligo cloning into DR274 are listed in Supplementary Table 4,5.

### Generation of p5E entry clones.
The NBT_tauGFP plasmid was a kind gift by Enrique Amaya. The 3.8 kb regulatory element derived from neural specific beta tubulin was removed from the NBT_tauGFP using SalI and HindIII and subcloned into SalI and HindIII digested and dephosphorylated p5E_MCS (ref. 64). The 1 kb flt1 enhancer/promoter fragment from the pMiniTol2_flt1_ECR5a_pro_181_YFP (ref. 39) construct was subcloned into p5E_MCS using KpnI and HindIII. The resulting plasmids were named p5E_ Xla.Tubb-3.8 and p5E_flt1$^{enh}$.

### Generation of a universal p2A-GFP middle entry clone.
To easily detect transgenic cells the pME_eGFP (#455) from the Tol2kit (ref. 64) was modified by site-directed mutagenesis PCR. The p2A sequence was added before the stop codon of GFP using pME_eGFP specific primer with 5′end extension coding for the p2A peptide and a SmaI restriction site just downstream of p2A for convenient subcloning (pME_eGFP_p2A_fw and pME_eGFP_p2A_rev primer).

### pME entry clones used for gain-of-function experiments.
pME_eGFP-p2A_SmaI was digested with SmaI and XhoI. The inserts vegfaa165, vegfC and sflt1 were amplified from zebrafish cDNA using primers vegfaa_p2A_fw/rev, vegfc_-p2A_fw/rev and sflt1_p2A_fw/rev. The PCR products were digested with XhoI and gel purified. Vector and inserts were ligated following the manufactures instructions (NEB T4 DNA Ligase). The resulting plasmids were named pME_eGFP-p2A_vegfaa165, pME_eGFP-p2A_vegfc and pME_eGFP-p2A_sflt1.

### gal4ERt2 middle entry clone generation.
To spatially and temporally regulate transgene expression an inducible gal4-ERT2 fusion protein was constructed. The Gal4 DNA binding domain was fused at its C-terminus with a mutant oestrogen ligand-binding domain ERT2 that carries a VP16-derived non-deleterious transactivation domain TA4 (ta4, 39 aa) at its C-terminus[65]. Among all possible sequential orders of domains, this arrangement was inferred to have a low background with a high induction rate[66]. A middle entry clone pENTR/D-creERT2 was modified by replacing Cre recombinase domain (1,053 bp, flanked by NotI and XhoI sites at its 5′ and 3′ termini, respectively) with a PCR product encoding Gal4 DNA binding domain (1–146 aa) with Kozak sequence in the 5′ vicinity of the start codon. To replace the stop codon at the 3′ terminus of ERT2 domain with TA4 domain, a C-terminal half of a ERT2 domain (115–316 aa, flanked by in-frame NcoI at its 5′ terminus) was replaced with a PCR product encoding the C-terminal half of the ERT2 domain without a stop codon (115–315 aa) with in-frame AgeI site at its 3′ terminus. The AgeI site, and 3′ downstream EcoRI site were utilized to insert two synthetic double-stranded oligonucleotides encoding the TA4 domain and the stop codon.

### Generation of gateway expression clones.
pME_DN-MAML-eGFP was kindly provided by Caroline Burns[47]. p5E_flt1$^{enh}$, pME_DN-MAML-eGFP and p3E_polyA were recombined into pDestTol2CG2 according to the manufacturer's instructions (Thermo Fisher, LR Clonase II plus). The resulting plasmid was named pCG2_flt1_ DN-MAML-eGFP. p5E_Xla.Tubb-3.8, pME_eGFP-p2A_sflt1 and p3E_polyA were recombined into pDestTol2CG2 (pCG2_Xla.Tubb-3.8_eGFP-p2a-sflt1). p5E_Xla.Tubb-3.8, pME_eGFP-p2A_sflt1

and *p3E_polyA* were recombined into pDestTol2CG2 (pCG2_Xla.Tubb-3.8_eGFP-p2A-sflt1). *p5E_elavl-3.2* (R.W., unpublished observations), *pME_gal4ERT2* and *p3E_polyA* were recombined into pDestTol2CG2 (pCG2_elavl-3.2_gal4-ERT2). *p5E_flt1^enh*, *pME_gal4ERT2* and *p3E_polyA* were recombined into pDestTol2CG2 (pCG2_ flt1^enh_gal4-ERT2).

**Generation of tissue-specific KO constructs.** *pME-Cas9-T2A-GFP* and *pDestTol2pA2-U6:gRNA* were a gift from Leonard Zon (Addgene plasmid # 63156 and # 63155)[49]. *pDestTol2pA2-U6:gRNA^flt1E3* was generated by annealed oligo cloning. Oligos U6_flt1E3_1 and U6_flt1E3_2 were cloned into *pDestTol2pA2-U6:gRNA* following BseRI restriction digest. To drive *Cas9* expression specifically in neurons, the Gal4 driver construct *pCG2_Xla:Tubb-3.8_gal4ERT2* was generated by recombining *p5E_Xla.Tubb-3.8*, *pME_gal4ERT2*, *p3E_polyA* and *pDestTol2CG2*. To drive Cas9 expression specifically in endothelial cells, the Gal4 driver construct *pCG2_ flt1^enh_gal4ERT2* was generated by recombining *p5E_ flt1^enh*, *pME_gal4ERT2*, *p3E_polyA* and *pDestTol2CG2*. For the Gal4 effector construct, *p5E_UAS*, *pME_cas9-t2a-eGFP* and *p3E_polyA* were recombined into *pDestTol2pA2-U6:sgRNA^flt1E3* (pCG2_UAS_Cas9-t2A-eGFP_U6_gRNA^flt1E3).

**Tissue-specific *miR155-flt1-1-2-3* knockdown constructs.** *sflt1* 3′UTR-specific miRNAs were designed using the BLOCK-IT RNAi Designer website (https://rnaidesigner.thermofisher.com/rnaiexpress/). To enhance miRNA effectiveness three sflt1 3′UTR-specific target sites with miRNA155 backbone were cloned in series. A fragment containing the three multiplexed miRNAs were synthesized by Eurofins Genomics and cloned into *641-pMER-GFP-miR155empty* and *641-pMER-DsRED-miR155empty* using restriction enzymes BamHI and XhoI[50]. The target sites are listed in Supplementary Table 7. The expression construct with *Xla.Tubb* or *flt1^enh* promoter was cloned using gateway cloning. *p5E_Xla.Tubb3.8*, *641-pMER-GFP-miR155-sflt1-1-2-3* and *p3E_polyA* were recombined into pDestTol2CG2 (pCG2_Xla.Tubb_ GFP-miR155-sflt1-1-2-3). *p5E_flt1^enh*,*641-pMER-DsRed-miR155-sflt1-1-2-3* and *p3E_polyA* were recombined into pDestTol2CG2 (pCG2_flt1^enh_DsRed-miR155-sflt1-1-2-3).

**FACS.** Approximately 500 embryos *Tg(mnx1:GFP)^ml2*, *Tg(HuC:EGFP)^as8*, *Tg(Xla.Tubb:DsRed)^zf148* or vhl MO injected *Tg(Xla.Tubb:DsRed)^zf148* embryos were dechorionated at 24 hpf using pronase (0.5 mg/ml). Cells were dissociated using FACSMax as recommended by the manufacturer. *Tg(mnx1:GFP)^ml2*, *Tg(HuC:EGFP)^as8* embryos were dissociated and sorted at 24 hpf, control and vhl MO injected *Tg(Xla.Tubb:DsRed)^zf148* embryos were dissociated and sorted at 3 dpf. Dissociated cells were FACS sorted using BD-FACS-Aria I and Aria II. The sorted cells (∼0.5 × 10^6 cells per experiment) were spun down at 310 g for 5 min and resuspended in lysis buffer contained in the RNeasy mini kit (Qiagen). RNA was extracted as described in the manual. Because of limited amounts of RNA the QuantiTect Whole Transcriptome Kit (Qiagen) was used to preamplify and reverse transcribe the RNA to make cDNA. cDNA was diluted 1:250 for real-time qPCR.

**Gene expression analysis by real-time qPCR and TaqMan.** Total RNA of zebrafish embryos was isolated with TRIzol, purified with RNeasy mini kit (Qiagen) and quantity and quality were measured using an Agilent 2,100 Bioanalyzer (Agilent Technologies) according to the manufacturer's instructions. We performed DNase on-column digestion using RNase-free DNase Set (Qiagen) according to the manufacturer, followed by cDNA synthesis using the Thermoscript First-Strand Synthesis System (Thermo Fisher Scientific). Primer probe sets (FAM and TAMRA labels) were obtained from Thermo Fisher Scientific. Amplification was carried out using an ABI Prism 7,000 thermocycler (Applied Biosystems). qPCR was conducted with SYBR Green PCR Master Mix (Thermo Scientific) in a StepOnePlus real-time qPCR system (Applied Biosystems). Primers for real-time qPCR were ordered from Eurofins Genomics. Gene expression data were normalized against zebrafish elongation factor 1-alpha. Primers and probes are listed in Supplementary Table 1–3.

**RNA-seq library preparation and sequencing.** Zebrafish RNA was isolated and purified from 4 dpf zebrafish larvae using TRIzol and RNeasy mini kit (Qiagen) as recommended by the manufacturers. A cDNA library was generated using the TruSeq Ilumina RNA sample prepv2 kit according to the manufacturer's protocol. The cDNA library was sequenced on a HiSeq2000 according to the manufacturer's protocols (Illumina).

**Identification of differentially expressed genes.** Raw sequencing reads were mapped to the transcriptome and the zebrafish reference genome (GRCz10 danRer10) using Bowtie2.0 and TopHat 2.0 (ref. 67). On average 44,490,573 reads (81,6% of total reads) were assigned to genes with Cufflinks and HTSeq software package. Differentially expressed genes (control vs. mutant) were identified using DESeq and Cuffdiff[67,68]. Genes were defined as differentially expressed if ≥2 fold

significantly regulated (P<0.05) with two independent methods (DEseq and Cuffdiff).

**Zebrafish histological sectioning.** Dechorionated larvae were fixed in 4%PFA for 2 h and subsequently transferred to 20% DMSO/ 80% Methanol and incubated overnight at −20 °C. Larvae were then washed in 100 mM NaCl, 100mMTris-HCl, pH7.4 for 30 min at room temperature. Washed larvae were embedded in gelatin from cold water fish skin/sucrose (Sigma). Larvae were sectioned (20 μm) in a cryomicrotome.

**Inhibitor treatments.** All stock solutions were prepared in DMSO. Embryos were dechorionated at 24 hpf using Pronase (Roche, Basel, Switzerland). For Notch signalling inhibition embryos were incubated from 2 dpf with 10 μM of LY-411575 (Sigma, St Louis, MO, USA) and imaged at 3 dpf. For VEGFR2 and VEGFR3 inhibition embryos were treated with 25 μM MAZ51 (Merck Millipore, Billerica, Massachusetts, USA) from 2.5 dpf or from 3 dpf with 0.125 μM ki8751 (Sigma, St Louis, MO, USA) and imaged at 4 dpf. To inhibit PI3K/Akt signalling embryos were incubated with 1.25 μM wortmannin from 3 dpf and imaged at analysed at 4 dpf. Heartbeat was blocked using 15 mM 2,3-Butanedione 2-monoxime (BDM) dissolved in E3 media. Control embryos were mock treated with DMSO (Sigma, St Louis, MO, USA). Embryos were randomly assigned to experimental groups. Investigators were blinded to inhibitor treatment.

**Photoconversion of kikGR and migration tracking.** Dechorionated embryos were embedded in 0.7% low-melting agarose at 30 hpf and a small part of the posterior cardinal vein of *Tg(kdrl:nlskikGR)^hsc7* transgenics was converted for several seconds using UV-light with the smallest available field diaphragm of the Leica Sp8 confocal microscope. Subsequently embryos removed from the agarose and allowed to develop in E3 medium until imaging or were immediately used for time-lapse imaging.

**Gal4ERT2 endoxifen activation.** Endoxifen (Sigma) was solved in DMSO. Zebrafish embryos expressing Gal4ERT2 were incubated from 52 hpf onwards in 0.5 μM endoxifen in E3 medium in the dark. GFP positive cells could be observed approximately 1.5 h after induction.

**Vascular network analysis.** To assess sprout number and length, we developed a semi-automated analysis of the DLAV-ISV vessel network using ImageJ (Supplementary Fig. 1n). Image-stacks of ISVs were acquired using the Leica SP8 confocal microscope. Stack projections of one side of the trunk were generated. Dorsal region of the ISVs was used for analysis. Using ImageJ a Gaussian blur filter was applied followed by a black/white threshold and subsequent skeletonization to generate a skeleton of the vasculature. Segment number, branch point number and total branch length were calculated using the 'analyse skeleton' plugin. The semi-automated pipeline was applied for analysis of 4 dpf vascular networks, while sprout numbers in 2–3 dpf zebrafish embryos were counted manually.

**Imaging.** Zebrafish larvae were embedded in 0.7% low-melting agarose with 0.112 mg ml^−1 Tricaine (E10521, Sigma) and 0.003% PTU (P7629, Sigma) in glass bottom dishes (MatTek, P35G-0.170-14-C). Images presented in this study were acquired using a Leica SP8 confocal microscope with ×20 multi-immersion and ×40 water immersion objectives and LAS X software. Images were processed using ImageJ. Vascular branching was quantified using a semi-automated ImageJ pipeline (Supplementary Fig. 1n). Animal numbers used are indicated in figure legends. For zebrafish mutants more than 100 embryos per genotype were analysed. In morpholino experiments morphologically malformed embryos were excluded from analysis.

**Statistical analysis.** Statistical analysis was performed using GraphPad Prism 6. Each dataset was tested for normal distribution (D'Agostino and Pearson test). Parametric method (unpaired Students *t*-test) was only applied if the data were normally distributed. For non-normal distributed data sets, a non-parametric test (Mann Whitney *U* test) was applied. When appropriate in case of multiple comparisons, ANOVA plus Bonferroni correction was applied. *P* values <0.05 were considered significant. Data are represented as mean ± s.e.m., unless otherwise indicated. *P<0.05, **P<0.01 and ***P<0.001.

**Data availability.** The authors declare that all data supporting the findings of this study are available within the article and its Supplementary Information files or from the corresponding author on reasonable request. The RNA-seq data generated in this study has been deposited into the Gene Expression Omnibus database with the accession code http://www.ncbi.nlm.nih.gov/geo/query/acc.cgi?acc=GSE89350.

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

## Acknowledgements

We thank the colleagues of the KIT–European Zebrafish Resource Center (EZRC) for handling and maintenance of the zebrafish lines. We are very grateful to Dr Caroline Burns, Cardiovascular Research Center, Charlestown for sharing the pME_DN-MAML-GFP entry clone. We thank Dr Thomas Becker and Dr Jean Giacomotto, Brain and Mind Research Institute, University of Sydney for the 641-pMER-GFP/DsRed-miR155empty plasmids. We thank Leonard I. Zon M.D., Boston Children's Hospital and Dana Farber Cancer Institute, Boston for the pDestTol2CG2-U6:gRNA and the pME-Cas9-T2A-GFP constructs and Dr Keith Joung, Massachusetts General Hospital, Charlestown for DR274 and MLM3613 plasmids. We are also very grateful to Dr Jeroen den Hertog, Hubrecht Institute, Utrecht for sharing the zebrafish *ptena*$^{-/-}$;*ptenb*$^{-/-}$ double mutants. S.S.-M., U.S. and F.L.N. are members of the EuFishBioMed zebrafish initiative. S.S.-M. and F.l.N. are supported by grants from the Deutsche Forschungsgemeinschaft (DFG)–FOR2325 'Interactions at the Neurovascular Interface'. Y.H. is supported by an individual grant from the Danish Council for Independent Research. We acknowledge support by Deutsche Forschungsgemeinschaft and Open Access Publishing Fund of Karlsruhe Institute of Technology.

## Author contributions

R.W. designed and performed experiments and interpreted experimental data. A.K. and R.W. performed and analysed FACS experiments. K.A. participated in manuscript preparation. L.P. performed and analysed inhibitor experiments. J.K. performed and analysed Taqman experiments. M.T. & U.S. contributed the Gal4ERT2 construct, Y.H. performed FACS analyses, K.A. & N.M. performed the analysis on pericyte contribution. A.van I. and S.S.-M. contributed transgenic fish, constructs, interpreted data and discussed the conceptual framework. F.l.N. conceived and designed the project, analysed the data and supervised the overall project. F.l.N. and R.W. wrote the manuscript.
