## [Peer Review File · Nature Communications]

Reviewer #1 (Remarks to the Author)

Wild and co-workers found that Vegf receptors and ligands are expressed at the neurovascular interface. Flt1 encodes a soluble, truncated protein, sFlt1, and a full length transmembrane protein, mFlt1. Flt1ka601, flt1ka602 and flt1ka603 mutants with a frameshift mutation in exon 3, which encodes Ig domain 1, display hyperbranching at the level of the neural tube. Flt1ka605 mutants with a frameshift mutation in exon 11b did not display any vascular defects. The ectopic sprouts in flt1ka601 mutants display distinctive properties in that the 55% of the sprouts from venous intersegmental vessel (vISV) that did make a connection, connected with arterial ISVs in more than 95% of the cases. In more than 80% of the cases, nuclear positioning was directly linked with sprout initiation. At 4 dpf, the number of endothelial cells is significantly increased in the vISV of flt1ka601 embryos. Ectopic venous sprouting was also observed in other models with enhanced expression of vegfaa, such as vhl mutants, ptena/ptenb double mutants, and embryos with neuronal specific expression of vegfaa. Real-time PCR indicated that neuronal cells express much higher levels of vegfaa than non-neuronal cells. Ectopic expression of sFlt1 rescued the hyperbranching defect in flt1ka601 mutants and so did morpholino-mediated knockdown of vegfaa and a KdrI inhibitor (R2 inhibitor). The authors conclude that they have identified a third sprouting mode, involving venularized arterial endothelium with a unique Vegfaa sensitivity and angiogenic potential enabling precise spatio-temporal control of neurovascular development.

This is an interesting paper that shows convincingly that elevated expression or bioavailability of vegfaa in neuronal cells induced hyperbranching at the level of the neural tube in developing zebrafish embryos. Overall, the data appear solid and the conclusions are largely supported by the data.

Points:

1. p.7, Suppl Fig. 2. Mutants ka601-603 have insertions/deletions in exon 3, resulting in severely truncated sFlt1 and mFlt1 proteins that are likely not to express any Flt1 protein. Ka605-608 mutants harbor mutations in exon 11b, resulting in truncated mFlt1 and presumably unaffected sFlt1. Is flt1 mRNA expressed normally in all these mutants, i.e. is there no nonsense-mediated decay? Is sFlt1 protein detected in ka605-608 mutants (and lost in ka601-603 mutants)? The authors conclude that the hyperbranching phenotype is caused by loss of sFlt1. If at all possible, showing Flt1 protein levels in the mutants would greatly strengthen the claim that loss of sFlt1 causes the hyperbranching phenotype. A minor point associated with Suppl. Fig. 2: What do the insertions/ deletions of the ka606 and ka608 mutants look like? These are mentioned in the text, but are not depicted in Supp Fig. 2.
2. p.9, Suppl. Fig. 4. "Key regulators of sprouting angiogenesis and markers of tip-stalk cells, including dll4, notch1a, hey/hes, ephrinb2a and nrarp were not altered". From the text it appears these data are provided in Supp. Fig. 4, but they are not. This should be explained.
3. Fig. 4h. The ratio of corrected total cell fluorescence was determined between vISV and aISV in Tg(kdrI:nlskikGR)hsc7 embryos upon photo-conversion. Based on these data, the authors state (p.16): "We furthermore show that after AV remodeling, segmental veins express more KdrI receptors than arteries, which may contribute to the high Vegfaa responsiveness in this domain". This is an overinterpretation of the data, given that the ratio vISV/aISV is only modestly enhanced (between 1.1 and 1.2). Why is photoconversion needed prior to quantification of fluorescence? Did the authors correct for the number of cells? This is not trivial, because endothelial cell proliferation is enhanced in vISVs of the mutant compared to wild type (Fig. 3k-I'). The transgenic line provides insight into kdrI-promoter activity, not KdrI protein levels on the target cells, which makes it very hard to conclude that veins will be more responsive to Vegfaa than arteries.
4. The authors conclude they have discovered a third sprouting mode (abstract). Whereas conceptually, they make a strong case that the observed sprouting in embryos with elevated levels of Vegfaa is distinct from previously described arterial and venous sprouting, it is not evident that the third mode of sprouting actually occurs during normal embryonic development. Could this third sprouting mode have a role in tumor vascularization?

Reviewer #2 (Remarks to the Author)

The manuscript by Wild et al is a showcase of currently available techniques in zebrafish research, and elegantly describes the role of sFlt1 expressing neurons in the formation of DLAV in zebrafish embryos. The data presented in the manuscript is of an exceptional quality and effectively supports the authors' claims. In addition, the authors findings are highly novel and interesting. However, a number of issues need to be resolved prior to the publication.

Main issues:

1. The wording in the abstract is rather strong. The authors claim that Flt1/VEGFAA regulated sprouting is the third mode. However, Flt1/VEGFAA appears to promote angiogenesis via indirectly activating Kdr1 signaling. Therefore, it is unclear whether this is a distinct mode of sprouting, or a simply a novel way to modulate Kdr1 signaling-induced sprouting angiogenesis.
2. In figure 1, the authors showed the relative mRNA expression levels. However, it is not clear how much indeed these mRNAs are expressed in the neuron. Having absolute value may help.
3. The authors utilizes Vegf165 exclusively. Is there any reason not to use other Vegf-As?
4. In figure 3, since transgenic lines are exclusively used to determine the expression of venous markers, it would be nice to show the endogenous expression of venous markers by in situ hybridization to confirm the correlation.
5. The section describes the upregulation of angiogenic sprout markers (pg 9), is rather inconclusive. At least the authors need to expand this section to explain what is the meaning of this finding. Since the majority of the sprouting angiogenic markers are not upregulated, it is not clear whether flt1 manipulation influences the expression of sprouting markers. Moreover, tip/stalk fate is dynamic, therefore, it would be difficult to assess the effects of flt1 on tip/stalk markers by microarray.
6. In figure 6, the authors elegantly show that neuron specific deletion of sFlt1 is driving the hyperbranching of the venularized arterial ECs. Maybe it would be technically challenging, but it would be nice to include EC specific deletion of sFlt1 as a comparison.
7. The interaction between neurons and vessels have been reported previously. It would be nice to introduce previous findings and describe how the new findings by the authors can expand our current understanding.

Reviewer #3 (Remarks to the Author)

Wild R. et al. demonstrate the significance of Flt1 expression in neural tube for regulating spinal cord vascularization. This study is an extension of their previous work (Kueger J et al. Development 138: 2111-2120, 2011) and aims at investigating the role for sFlt1 at the neurovascular interface. The authors clearly show that neurons function as a source of Vegfaa for pro-angiogenesis as well as a source of sFlt1 for anti-angiogenesis using several genetic mutants. The data was convincingly shown and deduced from the experiments executed logically to test their hypotheses. Yet, the conclusions might be biased by their hypotheses. Although there is no doubt about the importance of Vegfaa and sFlt1 in neural tube according to the present data, they might mislead readers to the wrong interpretation and subsequent conclusions. Thus, to obtain the correct conclusions, the authors are encouraged to address the following points and to interpret

their data more logically.

The points that should be clarified.

1. It is still unclear why only endothelial cells (ECs) expressing Flt4 (those in the secondary sprouts from the PCV) respond to Vegfaa from neural tube. The requirement of Kdr1 for hyper-branching in *flt1ka601* and/or *vhlhu2114* is clearly shown in the present study; however, it remains elusive why ECs of aISV do not respond to Vegfaa, although the ECs of aISV do express Kdr1. The ECs in vISV express highly sensitive Kdr1 and might potentially respond to Vegfaa. As the authors describe in the title, why do only Flt4- plus Kdr1-expressing ECs instead of those expressing only Kdr1 have high angiogenic potential?

The coverage of aISV by mural cells must inhibit the sprouting from aISV. This might account for the difference of abnormal sprouts from aISV and vISV. The authors need to explain the cause of this difference by performing additional experiments. The difference of Flt4 and Kdr1 expression is not enough to speculate the cause of hyper-branching of vISVs by using the words "endothelial cells with high angiogenic potential".

2. The authors seem to describe that the ECs of the dorsal region of vISV are venularized arterial ECs. Do they want to claim that the pre-existing ECs of the dorsal part of aISV before connecting to the ECs from the secondary sprout from the PCV become venous endothelial cells by changing their characters from arterial cells to venous endothelial cells? Flt4-expressing EC of the secondary sprout must migrate into the DLAV. Therefore, they need to explore whether the pre-existing arterial ECs change into venous ECs or those are pushed back toward the DLAV by the dorsally migrating Flt4-expressing venous EC of the secondary sprouts.

2-1. Flt4-positive ECs of the secondary sprout can be monitored by Flt4 promoter-driven fluorescence-expressing transgenic fish. If the authors carefully look at the Flt4-expressing cells, they might notice that those cells migrate into the DLAV, suggesting that the dorsal part of the ECs of pre-existing ISV (arterial ECs) do not change their character to venous ECs.

2-2. The authors can use the *Tg(kdr1:nlskikGR)* line to track the ECs in secondary sprout to test whether the ECs of secondary sprouts migrate into DLAV.

3. In relation to #2, if Vegfaa and sFlt1 from the neurons affect the ECs of the secondary sprouts, the cells constituting DLAV must be changed from the pre-existing arterial ECs to mixed population of ECs consisting of the pre-existing ECs and migrating ECs of secondary sprouts. In the *Flt1* mutant embryos, the number of ECs of DLAV must be increased in addition to hyper-branching of vISV. In Figure 3I and 3I', the cell number of vISV was counted. Similarly, the cell number of DLAV should be counted. Ideally, the number of Flt4-positive ECs originating from the secondary sprouts in DLAV should be counted. Indeed, the width of DLAV in wild type embryos appears to be less than that of *flt1ka601* mutant (Figure 2e' and 2g').

4. Figure 1c and supplementary movie 1 clearly show the indenting of not only vISV but also aISV, suggesting that Vegfaa and sFlt1 might determine the route of extension of primary sprouts. However, the primary sprouts from the dorsal aorta and the formation of DLAV were unaffected in the *Flt1* mutant fish. Why does this happen? The location of ISVs and neural tube (like Figure 1c) at early time points (initial blood vessel formation from primary sprouts to formation of DLAV before the connection to the secondary sprout) in both wild type and *Flt1* mutants should be analyzed carefully to examine the effects of Vegfaa and sFlt1 from neural tube on migration of ECs of primary sprouts. If the authors find indenting of ISV of primary sprouts, it is unclear why only ECs of vISV were affected in *Flt1* mutants (hyper-branched ECs are only in vISVs)? Because this point might puzzle the readers, the authors need to demonstrate the cause of difference of the effect of neural tube-derived molecules on primary sprouts and secondary sprouts. If the authors find the primary ISVs indenting neural tube, how is neuro-vascular interfaces of primary ISVs; the lateral interface between neural tube and ISVs and the dorsal interface between neural tube and DLAV, are regulated?

5. If the points #1-#4 are addressed, the main message of this study might be changed. Vegfaa

and sFlt1 might regulates not only the spatial patterning of vISV but also the DLAV formation. Accordingly, Figure 7 needs to be revised.

Minor points

- (1) Purple is not appropriate for the line. Other type of white broken line should be used in Figure 1e.
- (2) 30hpf in Figure 1 and 26(space)hpf in Figure 2. This expression should be used with consistency.

Reviewer #4 (Remarks to the Author)

Summary:

In their manuscript "Neuronal Flt1 controls spinal cord vascularization involving venularized arterial endothelium with high angiogenic potential." Wild and colleagues aim to study the molecular cross-talk between developing tissue and the endothelium. By using Zebrafish as an in vivo model they claim that developing spinal cord neurons coordinate endothelial cells proliferation at the neurovascular interface by titrating/buffering the local bioavailability of neuron-derived Vegfaa through expression of sFlt1. Mechanistically, they claim that veins at the neurovascular interface have higher expression of Kdr1 receptors enabling them to have a greater sprouting potential in response to Vegfaa than arteries. It is proposed that such mechanism allows neurons to fine-tune neuro-vascular development.

General comment:

This paper aims to characterize the formation of organ-specific vasculature by studying the cross-talk between developing tissue and the endothelium. A combination of both in-vivo and in-vitro assays is used to characterize the molecular mechanisms on how neuro-derived sFlt1 controls the bioavailability of Vegfaa to regulate angiogenesis at the neurovascular interface.

The quality and depth of analysis of the in-vivo data is remarkable. On the other hand, the mechanistic in-vitro data is not so compelling and needs further validation. Specifically, the claim that venularized arterial endothelial cells have a unique angiogenic potential due to a higher expression of the Kdr1 receptor remains elusive.

Overall, the manuscript is nicely written and presented, uses state of the art genetic techniques but tackles a very well characterized biological phenomenon (VEGF-sFLT1 antagonism). The conceptual novelty is, thus, somewhat limited.

Bellow, please find several comments, which are listed in the order of their appearance in the manuscript.

Specific comments - major:

1. Figure 1 (h) & (i)- Regarding the quantification of the mRNA expression levels, it is not clear how this data is presented. Does F.C. means Fold-change? The relative mRNA expression is relative to which sample? Authors should include this information on the legend of the Figure. Additionally, authors should present data as in Figure 5(b): mRNA expression levels of these genes should be compared to non-neuronal cells (GFP-negative cells). mRNA quantification by qPCR for a neuronal specific gene (when comparing GFP positive vs. GFP-negative cells) should also be included to demonstrate the purity of the sorted GFP-positive population.
2. The author`s claim that "two different neuronal reporter lines showed expression of mflt1, sflt1, kdr1, kdr, flt4 and the ligands vegfaa, vegfab, and plgf". When comparing the results from the two neuronal reporters used, the relative mRNA expression of sflt1 is shown to be down-regulated in HuC+ neurons and up-regulated in mnx1+ neurons. The same goes for other genes analyzed. What is the biological relevance of this finding? Could this be related to the purity of the sorted population?

3. Figure 2 - The author's claim that "The vascular phenotype observed in the *flt1ka601* mutants thus most likely involved soluble Flt1." - Why did the authors not generate a specific sFlt1 mutant by targeting exon E11a? This appears to be the most direct way to support author's findings.

4. Figure 4(h)- The author's claim that "Intersegmental veins express more *kdrl* receptor than arteries". The method used to support such finding is indirect and the small differences reported may not have biological relevance. Thus, more experimental evidence is needed to support the concept that veins at the neurovascular interface have a greater sprouting potential in response to Vegfaa than arteries due to higher expression of *Kdrl* receptors.

Specific comments - minor:

1. The title of the manuscript should emphasize soluble Flt1 as the major driver for the proposed mechanism.

2. Figure 1 legend - incorrect figure identification. (k) should be replaced by (j).

3. Figure 3 (m) - *Tdtomoto* should be replaced by *tdtomato*. Also, the concentration of MO that was used in this experiment is not clear. Please include this information in the figure legend.

Response to the reviewers:

We would like to thank the reviewers for their time, effort and valuable suggestions, which have allowed us to improve the manuscript substantially. While a detailed point-by-point response to all questions is provided below, we would like to start by highlighting the most important changes and additions:

1. We performed cell transplantation experiments. We find that transplantation of *flt1* mutant neurons into WT hosts induced ectopic sprouting. Transplantation of *flt1* mutant endothelial cells into WT hosts failed to induce sprouting (Fig. 8g-i). These data confirm that neuronal *flt1*, not vascular *flt1* controls sprouting angiogenesis in our setting.
2. To confirm the contribution of neuronal *sflt1* at the genetic level we introduced an additional genetic model. We find that neuron specific loss of *sflt1* induced sprouting (Supplementary Fig. 8b,c). Targeting vascular *sflt1* had no effect (Supplementary Fig. 8d). This shows that in neurons, expression of the soluble *flt1* isoform (*sflt1*) is functionally relevant.
3. We performed extensive cell tracking studies to establish the origin of the endothelial cells that gave rise to ectopic sprouts in the dorsal part of vISV upon loss of *flt1*. We show that not only remodeled artery derived cells, but also posterior cardinal vein (PCV) derived cells can colonize the dorsal part of vISV and contribute to ectopic sprouting (Fig. 4). We furthermore provide evidence showing that artery derived endothelial cells in venous segmental vessels, adapted their character to venous EC.
4. We clarified the mechanism accounting for the AV sprouting differences in *flt1* mutants. We show that arterial-venous differences in Notch signaling status account for the sprouting differences (Fig. 6a-f; Fig. 5f,g).
5. In response to reviewer 3, we addressed the potential contribution of mural cells-pericytes in AV sprouting differences. We now imaged pericytes in the embryonic trunk vasculature using a novel pericyte reporter line. We find that pericytes are rare during the time-window when sprouts arise and conclude that pericytes are not the cause for the AV sprouting differences, see point 4 (Fig. 6g-i).

Taken together: our data support the concept that neuronal sFlt1 restricts angiogenesis at the neuro-vascular interface.

Point-by-Point response to the reviewers

Reviewer #1.

Comment: Wild and co-workers found that Vegf receptors and ligands are expressed at the neurovascular interface. Flt1 encodes a soluble, truncated protein, sFlt1, and a full length transmembrane protein, mFlt1. Flt1ka601, flt1ka602 and flt1ka603 mutants with a frameshift mutation in exon 3, which encodes Ig domain 1, display hyperbranching at the level of the neural tube. Flt1ka605 mutants with a frameshift mutation in exon 11b did not display any vascular defects. The ectopic sprouts in flt1ka601 mutants display distinctive properties in that the 55% of the sprouts from venous intersegmental vessel (vISV) that did make a connection, connected with arterial ISVs in more than 95% of the cases. In more than 80% of the cases, nuclear positioning was directly linked with sprout initiation. At 4 dpf, the number of endothelial cells is significantly increased in the vISV of flt1ka601 embryos. Ectopic venous sprouting was also observed in other models with enhanced expression of vegfaa, such as vhl mutants, ptena/ptenb double mutants, and embryos with neuronal specific expression of vegfaa. Real-time PCR indicated that neuronal cells express much higher levels of vegfaa than non-neuronal cells. Ectopic expression of sFlt1 rescued the hyperbranching defect in flt1ka601 mutants and so did morpholino-mediated knockdown of vegfaa and a Kdr1 inhibitor (R2 inhibitor). The authors conclude that they have identified a third sprouting mode, involving venularized arterial endothelium with a unique Vegfaa sensitivity and angiogenic potential enabling precise spatio-temporal control of neurovascular development.

This is an interesting paper that shows convincingly that elevated expression or bioavailability of vegfaa in neuronal cells induced hyperbranching at the level of the neural tube in developing zebrafish embryos. Overall, the data appear solid and the conclusions are largely supported by the data.

Response: we would like to thank reviewer 1 for the positive and constructive comments. The remark that overall “the data appear solid and the conclusions are largely supported by the data” are well taken. Reviewer 1 raised some concerns that we addressed in additional experiments, and analyses. In particular we analyzed non-sense mediated decay in *flt1* mutants, generated tissue specific *sflt1* loss of function embryos, and addressed the cause for the AV sprouting differences.

Comment: p.7, Suppl Fig. 2. Mutants ka601-603 have insertions/deletions in exon 3, resulting in severely truncated sFlt1 and mFlt1 proteins that are likely not to express any Flt1 protein. Ka605-608 mutants harbor mutations in exon 11b, resulting in truncated mFlt1 and presumably unaffected sFlt1. Is *flt1* mRNA expressed normally in all these mutants, i.e. is there no nonsense-mediated decay?

Response: in line with the reviewers' suggestion we measured *flt1* mRNA expression and potential non-sense mediated decay using the deep sequencing data obtained from the *flt1*^{ka601} mutant (Supplementary Fig. 3). We found no signs of non-sense mediated decay. Similar observations were made in *mflt1* mutants.

For analyzing the RNAseq data we generated a so-called sashimi plot (Supplementary Fig. 3a). In this plot, the *sflt1* and *mflt1* intron-exon structures are indicated in the bottom panel of the graph, and the number of corresponding reads for each exon presented in the upper panels. The number of reads per exon position is represented as peaks and the read numbers are indicated. Exon spanning reads are indicated by the arc symbol. For *mflt1* and *sflt1* – reads were detected in all exons and comparable between WT and *flt1*^{ka601} mutant. To substantiate these data, we in addition performed quantitative PCR for *sflt1* and *mflt1* in the *flt1*^{ka601} (full mutant) and the *flt1ka*⁶⁰⁵ mutant (*mflt1* specific mutant). In both mutants we observed expression of *sflt1* and *mflt1*, at levels comparable to WT. We conclude that there are no signs for non-sense mediated decay in *flt1* mutants.

Comment: Is sFlt1 protein detected in ka605-608 mutants (and lost in ka601-603). The authors conclude that the hyperbranching phenotype is caused by loss of sFlt1. If at all possible, showing Flt1 protein levels in the mutants would greatly strengthen the claim that loss of sFlt1 causes the hyperbranching phenotype.

Response: We tested a series of commercially available Flt1 antibodies but they unfortunately did not work in our samples. However to substantiate the claim that *sflt1* is involved we explored alternative genetic approaches. For this purpose we generated tissue specific *sflt1* loss of function embryos. For generating tissue specific *sflt1* loss of function embryos we used a recently developed miRNA based technique with high knockdown efficiency, optimized for use in zebrafish (Giacomotto et al. 2015). We employed multiple custom designed miRNAs directed against *sflt1* arranged in series with a common miR155 backbone (see materials and methods part “Generation of tissue-specific miR155-flt1-1-2-3 knockdown constructs” and (Giacomotto et al. 2015)). To obtain tissue specificity we cloned the expression constructs under control of vascular (*flt1*^{enh}) and neuronal (*Xla.Tubb*) specific promoters. We observed that targeting neuronal *sflt1* resulted in ectopic venous sprouting (Supplementary Fig. 8b,c). Targeting vascular *sflt1* did not induce hypersprouting (Supplementary Fig. 8b,d).

Comment: What do the insertions/ deletions of the ka606 and ka608 mutants look like? These are mentioned in the text, but are not depicted in Supp Fig. 2.

Response: We apologize for this omission. We now provide the requested insertion-deletion sequence data in Supplementary Fig. 2e.

Comment: p.9, Suppl. Fig. 4. "Key regulators of sprouting angiogenesis and markers of tip-stalk cells, including *dll4*, *notch1a*, *notch1b*, *hey/hes*, *ephrinb2a* and *nrarp* were not altered". From the text it appears these data are provided in Supp. Fig. 4, but they are not. This should be explained.

Response: We now present a new figure (Supplementary Fig. 5) showing the heat map of the RNAseq data (panel a) and added separate panels showing the expression of mentioned markers (panels b,c). The heat map of the RNAseq data is presented in Supplementary Fig. 5a. In Supplementary Fig. 5b we show the expression of *notch1a*, *notch1b*, *dll4*, *nrarpa*, *nrarpb*, *hey1*, *hey2*, *her6*, (all of which are not significantly deregulated) as well as *esm1*, *angpt2a* (which are deregulated) based on the RNAseq results presented in Supplementary Fig. 5a. We next verified upregulated expression of *esm1*, *angpt2a*, *aplra*, *lyve1* and Flt1 ligand *plgf* using qPCR (Supplementary Fig. 5c).

The reasoning behind our analysis is the following. We find that ectopic venous sprouting is driven by Vegfaa-Kdrl, and sprouting occurs relatively late, around 2.5dpf. In the classical tip-stalk cell model, Vegf-Kdr drives *dll4* in tip cells resulting in activation of Notch in stalk cells. In stalks, active Notch causes down regulation of *kdr*, *flt4*, and upregulation of *flt1* and *nrarp*, collectively reducing Vegf responsiveness and sprouting of stalk cells. In most settings hyper-sprouting is explained by loss of *dll4* or *notch*. However, in our setting we do not find evidence for deregulated *dll4* or *notch*. One explanation for these results is that in our setting, ectopic sprouts emanate from veins, and both *notch* and *dll4* are not expressed in veins. Thus in Vegfaa driven ectopic venous sprouting, Kdrl couples to angiogenic cell behavior without changes in Notch signaling status. In line with this we show that inhibition of Notch in *flt1* mutants using a gamma secretase inhibitor did not augment venous sprouting (the Notch inhibitor was applied after AV remodeling, 2dpf). Adding this gamma-secretase inhibitor to WT embryos at 2dpf, did not induce sprouting, suggesting that loss of *flt1* is needed to obtain ectopic sprouts during this stage of development.

Besides the classical markers (*dll4*, *notch*, *nrarp*), other genes have been identified that are enriched in sprouting vessels and can be used as biomarker for sprouting events. These include apelin, apelin receptor, angiopoietin2, and *esm1* (Strasser et al. 2010; del Toro et al. 2010; Rocha et al. 2014). These molecules are implied in regulation of tip-stalk cell orientation and cross-talk. We indeed find that these sprouting biomarkers are upregulated in the *flt1* mutants (Supplementary Fig. 5b,c). This is what we tried to emphasize in our results discussion.

Comment: Fig. 4h. The ratio of corrected total cell fluorescence was determined between vISV and aISV in Tg(*kdr1:nlskikGR*)*hsc7* embryos upon photo-conversion. Based on these data, the authors state (p.16): "We furthermore show that after AV remodeling, segmental veins express more Kdrl receptors than arteries, which may contribute to the high Vegfaa responsiveness in this domain". This is an overinterpretation of the data, given that the ratio vISV/aISV is only modestly enhanced (between 1.1 and 1.2). Why is photoconversion needed prior to quantification of fluorescence? Did the authors correct for the number of cells? This is not trivial, because endothelial cell proliferation is enhanced in vISVs of the mutant compared to wild type (Fig. 3k-l'). The transgenic line provides insight into *kdr1*-promoter activity, not Kdrl protein levels on the target cells, which makes it very hard to conclude that veins will be more responsive to Vegfaa than arteries.

Response: We agree with the reviewer that our statement was over enthusiastic, and based on the requested analyses and new experiments, removed this phrase. We now present new data addressing the cause of the arterial-venous sprouting differences (Fig. 6; Fig. 5f,g), see paragraph below.

Our reasoning to perform photoconversion was that during early development, the *kdr1* promoter is slightly more active in arteries when compared to veins, and GFP protein has a relatively long half-life. Therefore venularized arterial endothelial cells in vISV may contain GFP protein generated while these endothelial cells were being part of the aISV. We therefore decided to perform the photoconversion after the completion of the AV

remodeling process to minimize the potential contribution of fluorescent protein produced by arterial endothelial cells prior to ending up in vISV.

In line with the reviewers' suggestion we carefully quantified arterial and venous endothelial cells numbers in WT and *flt1* (Fig. 3k-m), and corrected GFP levels for endothelial cell number. Using this approach, we find that the vISV/aISV ratio shifts from 1.2 to below 1.0 suggesting that arteries may have higher *kdr1* promoter activity than veins. As the reviewer correctly indicated, our initial statement about higher *kdr1* in vISV, may thus not hold, as endothelial cell number differs. We adapted our manuscript accordingly. We therefore reconsidered other explanations for AV sprouting differences. In short: we find that Notch may play a more prominent role than originally anticipated.

It is well established that Notch acts as a negative regulator of sprouting angiogenesis, and Notch signaling is high in arteries and low in veins. We inhibited Notch in ISVs using an UAS driven *dnMAML* approach. In line with Notch acting as a repressor, we find that inhibiting Notch with *dnMAML* using the UAS driven approach resulted in ectopic arterial sprouting. Based on feed back from colleagues in the zebrafish community and a very recent paper published by the group of Lawson (Shin et al. 2016), our technical approach may have a technical limitation. These authors report that UAS driven approaches, result in mosaic and variable expression of the target gene. If this is correct, we cannot rule out that we did not achieve complete Notch inhibition with our *dnMAML* approach. Residual Notch activity may have contributed to repressing sprouting. To overcome this technical problem we now used a pharmacological approach to inhibit Notch (Stegmaier et al. 2014). We added the gamma secretase inhibitor LY-411575 at 2dpf (after the AV remodeling) to *flt1*^{ka601} mutants and observed ectopic arterial sprouting, at levels twice as high as observed with the *dnMAML* approach (Fig. 6c-f). The difference between LY-411575 and *dnMAML* indeed supports that there was residual Notch activity in the latter scenario.

Reviewer 1 pointed us toward investigating endothelial cell numbers in detail, and asked what happens when we correct the observed events for the number of cells. We substantiated the endothelial cell data (Fig. 3k-m). We observed a clear difference between cell numbers in aISV and vISVs; in *flt1* mutants, aISV contained on average 10 endothelial cells, whereas vISV contained 15 EC; thus in vISV more endothelial cells maybe available for sprouting events when compared to aISV.

Of note: we complemented our study with knockdown of *dll4* in *flt1*^{ka601}. Loss of *dll4* prior to AV remodeling, results in a trunk vasculature that consists almost exclusively of venous ISVs (Fig. 5f,g) as previously described (Leslie et al. 2007). This is attributed to Dll4-Notch's role in specification of arterial identity, and loss of Notch induces venous identity. Accordingly, in support of a role for venous endothelium, loss of *dll4* in *flt1* mutants augmented branching when compared to *flt1* mutants (Fig. 5f,g). Of course Notch is tightly coupled to both vessel identity and sprouting (Leslie et al. 2007; Quillien et al. 2014). Veins show low Notch and thus lack a repressor, which may favor vISV sprouting. We added these new data to our manuscript and adapted our conclusions accordingly.

Comment: it is not evident that the third mode of sprouting actually occurs during normal embryonic development. Could this third sprouting mode have a role in tumor vascularization?

Response: we obtained evidence for tertiary sprouting during normal development (Fig. 7g-l and Supplementary Fig. 7e-g). Since loss of *flt1* resulted in ectopic sprouting around the spinal cord we decided to investigate spinal cord vascularization of WT embryos in more detail. In WT, the spinal cord becomes vascularized in the period between 12-14dpf, involving sprouts derived from ISVs. Careful analysis shows that the sprouts preferentially emanated from venous ISV (92%). At the cellular level we furthermore provide evidence for nuclear positioning in these sprouts (Supplementary Fig. 7f). If indeed *flt1* is involved in normal spinal cord vascularization, we hypothesized that *flt1* levels go down during the period when spinal cord vascularization commences. We indeed found that *sflt1* is down regulated at 12dpf when compared to the early stage embryos (Supplementary Fig. 7g). We hypothesize that loss of neuronal *sflt1* around 12dpf may permit spinal cord vascularization.

The reviewer mentions sprouting in tumors, and we agree with the reviewer that from a medical therapeutic point of view, it would be very interesting to investigate this in more detail. It is worth mentioning that in the older angiogenesis literature (80's and early 90's) from Judah Folkman, Olga Hudlicka, and Werner Risau (Nugent & O'Connor 1983; Risau 1995), at the time leaders in the field, venous sprouting is often mentioned with regard to tumor vascularization, and physiological angiogenesis in skeletal muscle. In line with the reviewers' suggestion we are in the progress of developing brain tumor models in zebrafish to study the prevalence of venous sprouting in more detail. We believe however that these tumor data are outside of the scope of the current manuscript.

Reviewer #2.

Comment: The manuscript by Wild et al is a showcase of currently available techniques in zebrafish research, and elegantly describes the role of sFlt1 expressing neurons in the formation of DLAV in zebrafish embryos. The data presented in the manuscript is of an exceptional quality and effectively supports the authors' claims. In addition, the authors findings are highly novel and interesting. However, a number of issues need to be resolved prior to the publication.

Response: we would like to thank the reviewer for the positive comments and constructive feedback. In line with reviewers' comments, we have performed additional experiments and addressed the concerns.

Comment: The wording in the abstract is rather strong. The authors claim that Flt1/VEGFAA regulated sprouting is the third mode. However, Flt1/VEGFAA appears to promote angiogenesis via indirectly activating Kdr1 signaling. Therefore, it is unclear whether this is a distinct mode of sprouting, or a simply a novel way to modulate Kdr1 signaling-induced sprouting angiogenesis.

Response: we agree with the reviewer that the statement maybe strong and have therefore down tuned our statements and rewrote the abstract. Accordingly we removed the term tertiary sprouting from the manuscript. However, we do believe that our findings support both a distinct mode of sprouting, and a novel way to modulate Kdr1 signaling via neuronal sFlt1. To clarify the issue about similarities and differences between the primary, secondary and our loss of *flt1* induced sprouting mode, we provide a short summary of what is known.

In zebrafish trunk vascular development, primary artery sprouting and secondary venous sprouting are of critical importance to form a stable vascular network. It is established that primary artery sprouting is driven by Shh-Vegfaa-Kdr1-Dll4/Notch – PLCgamma-Erk signaling pathway. Primary artery sprouting can occur in the absence of blood flow perfusion. Characteristic for primary sprouting is the active tip-stalk cell shuffling, and the contribution of Dll4-Notch in tip-stalk cell differentiation. Secondary venous sprouting is driven by Vegfc-Flt4-PI3/Akt. Ectopic venous sprouting in *flt1* mutants however, is driven by Vegfaa-Kdr1. Ectopic venous sprouting involves nuclear positioning which has not been described for the other sprouting forms. Ectopic venous sprouting requires blood flow as loss of hemodynamics completely annihilates ectopic sprouting (Supplementary Fig. 9a,b). Ectopic venous sprouting did not involve tip stalk cell shuffling. In addition we find no evidence for altered expression of the classical tip-stalk markers *dll4*, *notch* or *nrarp*. These markers are considered downstream of Kdr1-Notch and deregulated in the case of hyperbranched arteries. We also explain why these genes were unaltered: ectopic venous sprouts emanate from veins, and veins do not express *dll4* or *notch*. Both secondary venous sprouts and ectopic venous sprouts preferentially anastomose with arteries.

Interestingly ectopic venous sprouting is coordinated by neurons, as loss of neuronal *flt1* promotes the formation of these sprouts. This regulation process differs from the role of *flt1* in the classical tip-stalk cell model where vascular Notch drives *flt1* in vascular stalk cells, thus reducing Vegf responsiveness and sprouting of stalks. Ectopic venous sprouts however emanate from veins, and Notch signaling is absent in

intersegmental veins. We furthermore provide cell transplantation experiments showing that transplantation of *flt1* mutant neuronal cells, not vascular cells, induced sprouting. We now also provide evidence showing that ectopic venous sprouting is not restricted to *flt1* mutants, but can be observed in older WT embryos contributing to spinal cord vascularization (Fig. 7g-i and Supplementary Fig. 7e-g).

Comment: In figure 1, the authors showed the relative mRNA expression levels. However, it is not clear how much indeed these mRNAs are expressed in the neuron. Having absolute value may help.

Response: we have redone and substantiated the quantitative PCRs on the FAC-sorted neuronal cells and now present a new figure with additional comparisons between *flt1* and neuronal guidance gene expression (Supplementary Fig. 1). To substantiate that neuronal *flt1* expression levels are indeed physiologically relevant we performed additional cell-transplantation experiments (Fig. 8g-i). We transplanted *flt1* mutant neuronal cells or *flt1* mutant endothelial cells into WT background. We find that transplantation of *flt1* mutant neuronal cells into WT resulted in ectopic venous sprouting. Transplantation of *flt1* mutant endothelial cells failed to induce sprouting. To confirm the contribution of neuronal *sflt1* we generated neuron specific *sflt1* loss of function embryos using a special miRNA approach targeting neuronal *sflt1* mRNA (Supplementary Fig. 8b,c; details on method in (Giacomotto et al. 2015)). We find that targeting neuronal *sflt1* induces ectopic sprouting (Supplementary Fig. 8c). Targeting vascular *sflt1* had not effect (Supplementary Fig. 8b,d). Based on these lines of evidence, and substantiated by the neuron specific CRISPR/Cas9 *flt1* mutant, and neuron specific *sflt1* rescue (Fig. 8a-f) we conclude that neurons express physiologically relevant *sflt1* levels controlling sprouting at the neurovascular interface.

Comment: The authors utilizes Vegf165 exclusively. Is there any reason not to use other Vegf-As?

Response: there was not particular reason for exclusively using vegfaa165. In line with the reviewers' suggestion we now present data for the Vegfaa121 isoform (see Supplementary Fig. 8g). We overexpressed *vegfaa121* under control of an inducible neuronal promoter, similar to the approach used for *vegfaa165*. We find that *vegfaa121* induced ectopic sprouting at the level of the neural tube and anastomosis formation.

Comment: In figure 3, since transgenic lines are exclusively used to determine the expression of venous markers, it would be nice to show the endogenous expression of venous markers by in situ hybridization to confirm the correlation.

Response: the *flt4* reporter line and corresponding *flt4* in situ's have been published previously (Hogan et al. 2009; Gordon et al. 2013). When we compare the *flt4* in situ's presented in the study by (Hogan et al. 2009), and the *flt4* in situ's presented in a recent study by the group of Lawson (Shin et al. 2016) with the *flt4* reporter, we conclude that the *flt4* reporter recapitulates the *flt4* in situ's. Of note: we do not want to claim that *flt4* exclusive marks veins. We used the *flt4* promoter reporter because it gives a much nicer cellular resolution when compared to in situ's, especially in the older embryos.

Comment: The section describes the upregulation of angiogenic sprout markers (pg 9), is rather inconclusive. At least the authors need to expand this section to explain what is the meaning of this finding. Since the majority of the sprouting angiogenic markers are not upregulated, it is not clear whether *flt1* manipulation influences the expression of sprouting markers. Moreover, tip/stalk fate is dynamic, therefore, it would be difficult to assess the effects of *flt1* on tip/stalk markers by microarray.

Response: we agree that some markers are deregulated whereas other genes are not. We explain why this (see below) and have expanded our results discussion. To clarify this issue we now present a new figure (Supplementary Fig. 5) showing the heat map of the RNAseq data (Supplementary Fig. 5a). Based on RNAseq we quantified the expression of *notch1a*, *notch1b*, *dll4*, *nrarpa*, *nrarpb*, *hey1*, *hey2*, *her6*, (all of which are not significantly deregulated) as well as *esm1*, *angpt2a* (which are deregulated) in Supplementary Fig. 5b. We next verified upregulated expression of *esm1*, *angpt2a*, *aplnra*, *lyve1* and Flt1 ligand *plgf* using qPCR; the results of this are depicted in Supplementary Fig. 5c.

The reasoning behind our analysis is the following. We show that ectopic venous sprouting is driven by Vegfaa-Kdr1. In the classical tip-stalk cell model, Vegf-Kdr drives *dll4* in tip cells resulting in activation of Notch in stalk cells. In stalks, active Notch causes down regulation of *kdr*, *flt4*, and upregulation of *flt1* and *nrarp*, collectively reducing Vegf responsiveness and sprouting of stalk cells. In most settings hyper-sprouting is explained by loss of *dll4* or *notch*. However, in our setting we do not find evidence for deregulated *dll4* or *notch*. Another difference with the classical tip-stalk cell model is that we do not find evidence for tip-stalk cell shuffling.

One explanation for these results is that in our setting, ectopic sprouts emanate from veins, and both *notch* and *dll4* are not expressed in veins (Quillien et al. 2014), thus limiting their use as tip/stalk marker in our setting. This leaves open the option that in Vegfaa driven ectopic venous sprouting, Kdr1 couples to angiogenic cell behavior without changes in Notch signaling status. In line with this we show that inhibition of Notch in *flt1* mutants using a gamma secretase inhibitor did not augment venous sprouting (the Notch inhibitor was applied after AV remodeling, 2dpf). Adding this gamma-secretase inhibitor to WT embryos at 2dpf, did not induce sprouting, suggesting that loss of *flt1* is needed to obtain ectopic sprouts during this stage of development. In contrast, inhibiting Notch during primary arterial sprouting (24hpf, prior to completion of AV remodeling) where Vegfaa-Kdr1 mediated sprouting is coupled to Dll4-Notch, affects sprouting behavior.

Besides the classical markers (*dll4*,*nrarp*), other genes have been identified that in some scenarios are enriched in sprouting vessels and can be used as biomarker for sprouting events. These include apelin, apelin receptor, angiopoietin2, and *esm1* (del Toro et al. 2010; Strasser et al. 2010). These molecules are implied in regulation of tip cell orientation and cross-talk (del Toro et al. 2010; Rocha et al. 2014). We indeed find that some of these sprouting biomarkers are upregulated in the *flt1* mutants (Supplementary Fig. 5b,c). This is what we tried to emphasize in our results discussion. Thus while *esm1* maybe observed in sprouts of loss of notch scenarios, it could well be that *esm1* is also a suitable biomarker for angiogenic events in cases where notch is not deregulated. Of note: most tip-stalk cell markers were identified by microarray analysis of isolated endothelial cell derived from post-natal mouse retinas (del Toro et al. 2010; Strasser et al. 2010). Instead we used RNAseq.

Comment: In figure 6, the authors elegantly show that neuron specific deletion of sFlt1 is driving the hyperbranching of the venularized arterial ECs. Maybe it would be technically challenging, but it would be nice to include EC specific deletion of sFlt1 as a comparison.

Response: we agree: to clarify the contribution of neuronal versus vascular *flt1* in mediating the sprouting phenotype, we performed cell transplantation experiments (see response on previous comment). We find that transplantation of *flt1* mutant neuronal cells into WT hosts induced sprouting, whereas transplantation of *flt1* mutant endothelial cells did not (Fig. 8g-i). In addition we also explored genetic approaches to strengthen the claim that sFlt1 causes hypersprouting. For this purpose we generated tissue specific *sflt1* loss of function embryos. For generating tissue specific *sflt1* loss of function embryos we used a recently developed miRNA based technique optimized for use in zebrafish (Giacomotto et al. 2015). We employed multiple custom designed miRNAs directed against *sflt1* arranged in series with a common miR155 backbone (see materials and methods part “Generation of tissue-specific miR155-*flt1*-1-2-3 knockdown constructs” and (Giacomotto et al. 2015)). To obtain tissue specificity we cloned the expression constructs under control of vascular (*flt1^{enh}*) and neuronal (*Xla.Tubb*) specific promoters. We observed that targeting neuronal *sflt1* resulted in ectopic venous sprouting (Supplementary Fig. 8b,c). Targeting vascular *sflt1* did not induce hypersprouting (Supplementary Fig. 8b,d). Finally, and in line with the reviewers’ suggestion we also targeted endothelial *flt1*, using a comparable CRISPR/Cas9 approach as for neurons. To genetically target endothelial *flt1*, we overexpressed Cas9 under control of the *flt1* enhancer promoter (*flt1^{enh}*), which is predominantly active in ISVs, and the corresponding *flt1* targeting sgRNAs was expressed ubiquitously. Cas9 expression was readily observed in ISVs however we did not find ectopic sprouting events (see Supplementary Fig. 8a). Taken together, based on the transplantation data and the genetic targeting approaches, we propose that neuronal *sflt1* is the physiologically relevant mediator.

Comment: The interaction between neurons and vessels have been reported previously. It would be nice to introduce previous findings and describe how the new findings by the authors can expand our current understanding.

Response: we agree, among the first to show nerve vessel interaction in the context of vascular development was Yoh-suke Mukoyama, at the time in the lab of David Anderson (Mukoyama et al. 2002). They showed that in the embryonic mouse skin, sensory nerves provide *Vegfa* to the blood vessel capillary plexus, promoting expression of arterial marker neuropilin-1 and arteriogenesis. In older studies it has been established that arteries secrete molecules that attract sympathetic nerves controlling perivascular innervation of resistance arteries, which is physiologically relevant for controlling organ perfusion and blood pressure. In line with the reviewers’ suggestion, we have expanded the part on neuro-vascular cross-talk (within the limits of the maximum word-count of the MS; for additional reviews we refer to (Mukoyama 2008; Ruhrberg & Bautsch 2013).

Reviewer #3.

Comment: Wild R. et al. demonstrate the significance of Flt1 expression in neural tube for regulating spinal cord vascularization. This study is an extension of their previous work (Krueger J et al. Development 138: 2111-2120, 2011) and aims at investigating the role for sFlt1 at the neurovascular interface. The authors clearly show that neurons function as a source of Vegfaa for pro-angiogenesis as well as a source of sFlt1 for anti-angiogenesis using several genetic mutants. The data was convincingly shown and deduced from the experiments executed logically to test their hypotheses. Yet, the conclusions might be biased by their hypotheses. Although there is no doubt about the importance of Vegfaa and sFlt1 in neural tube according to the present data, they might mislead readers to the wrong interpretation and subsequent conclusions. Thus, to obtain the correct conclusions, the authors are encouraged to address the following points and to interpret their data more logically.

Response: We would like to thank reviewer 3 for the constructive feedback on our work. The comment that “The data was convincingly shown and deduced from the experiments executed logically to test their hypotheses” is well taken. We are also grateful for the suggestions to improve our work. We have performed a substantial set of new experiments addressing the origin of the endothelial cells in the dorsal aspect of vISV, and the contribution to ectopic sprouting (new Fig. 4). We furthermore addressed the mechanism accounting for the AV sprouting differences, as well as the potential role of pericytes herein (Fig. 6). Finally, we performed endothelial and neuronal cell transplantation experiments (Fig. 8g-i), and we generated neuron specific *sflt1* loss of function embryos. Based on the new data and in line with the reviewers’ suggestion we corrected our conclusions, title, and adapted the manuscript accordingly. For clarity we would like to start with addressing the origin of the endothelial cells in vISV (point 2), followed by explanations for the AV sprouting differences (point 1), and the remaining points.

Origin of endothelial cells in vISV and ectopic sprouting:

Comment: point 2, The authors seem to describe that the ECs of the dorsal region of vISV are venularized arterial ECs. (a) Do they want to claim that the pre-existing ECs of the dorsal part of aISV before connecting to the ECs from the secondary sprout from the PCV become venous endothelial cells by changing their characters from arterial cells to venous endothelial cells? (b) Flt4-expressing EC of the secondary sprout must migrate into the DLAV. Therefore, they need to explore whether the (b) pre-existing arterial ECs change into venous ECs or those are pushed back toward the DLAV by the dorsally migrating Flt4-expressing venous EC of the secondary sprouts. 2-1. Flt4-positive ECs of the secondary sprout can be monitored by Flt4 promoter-driven fluorescence-expressing transgenic fish. If the authors carefully look at the Flt4-expressing cells, they might notice that those cells migrate into the DLAV, suggesting that the dorsal part of the ECs of pre-existing ISV (arterial ECs) do not change their character to venous ECs. 2-2. The authors can use the Tg(*kdr1:nlskikGR*) line to track the ECs in secondary sprout to test whether the ECs of secondary sprouts migrate into DLAV.

Response: reviewer 3 asked to examine the origin and movements of the endothelial cells in the dorsal aspect of vISV. We thank the reviewer for suggesting this experiment and the results indeed shed a new light on our data. As suggested we used the Tg(kdrl:nlskikGR) line to perform cell tracking experiments (Fig. 4; Supplementary movie 5). We photo-converted a small part of the PCV at 30hpf and tracked the endothelial migration events in the period 30-60hpf by time-lapse imaging (Fig. 4). We observed three scenarios.

In scenario 1: we find that PCV derived venous endothelial cells migrated into the vISV and reached the most dorsal aspect of the vISV (Fig. 4a''-a'''). In the most dorsal aspect we find that PCV derived endothelial cells co-existed with the (remodeled venularized) artery derived endothelial cells (Fig. 4b). Scenario 1, which we refer to as "mixed" (mixed: meaning containing both artery and vein derived endothelium), accounted for 45% of cases. In scenario 1: We next quantified the percentage of artery derived and vein derived endothelial cells in the mixed population, located in the dorsal part of vISV in more detail (Fig. 4c, right panel). In the mixed scenario, we find that 67.9% of endothelial cells were of venous origin, and 32.1% of arterial origin.

In scenario 2: we find that the PCV derived venous endothelial cells migrated into the dorsal aspect of vISV. In this scenario, artery derived endothelial cells were absent, and the region only contained venous derived EC. Scenario 2 accounted for 48.6% of cases (Fig. 4c). In scenario 3 we find vISV that were not colonized by migrating PCV derived venous endothelial cells, and the dorsal aspect of these vISV consisted of artery-derived endothelium only. This scenario accounted for 8.2% of cases (Fig. 4c). A graphical summary of these scenarios is presented in Fig. 4d.

We next asked if both artery derived and vein derived endothelial cells can give rise to ectopic sprout upon loss of *flt1*, and found that this is indeed the case (Fig. 4e-i). We found sprouts consisting of artery derived endothelium, and sprouts consisting of venous derived endothelium (Fig. 4e,f,h,h''). In addition, we observed composite sprouts containing both an artery and venous derived endothelial cell juxtapositioned (Fig. 4f,h').

Taken together: with respect to the origin of the endothelial cells in the dorsal part of vISV we find three scenarios: scenario 1) the dorsal part of vISV contains mixed artery- and vein-derived endothelial cell population, scenario 2) the dorsal part is exclusively populated by vein-derived cells, scenario 3) the dorsal part only contains artery-derived endothelial cells (Fig. 4d). We show that within vISV, both artery-derived and venous-derived endothelial cells can give rise to ectopic sprouts upon loss of *flt1* (Fig. 4h-h'').

We originally stated the ectopic sprouting is mediated by remodeled venularized arterial endothelium. Our new data show that ectopic sprouts can also emanate from PCV derived venous endothelium. Therefore our original statement is not correct, and in line with the reviewers' suggestion, we adapted the manuscript according to the new data.

The reviewer states that "the dorsal part of the ECs of pre-existing ISV (arterial ECs) do not change their character to venous ECs". Here, based on the new data sets, we show that this statement is only partly correct. Indeed in 48.6% of cases (scenario 2), we observed that the dorsal part of vISV consisted of venous-derived endothelial cells only. This suggests that, in these cases, the artery-derived endothelial cells were displaced from the vISV, most likely toward the DLAV. Here they may not have to adapt their arterial phenotype since Notch is active in the DLAV (as in aISV). However in the other cases (scenario 1+3), artery-derived endothelial cells remained in the dorsal part of vISV. We

subsequently showed that artery-derived endothelium in the dorsal part of vISV contributed to ectopic sprouting upon loss of *flt1*. Artery-derived endothelium in arterial ISVs never showed ectopic sprouting upon loss of *flt1*. Furthermore inhibiting AV remodeling by targeting *flt4*, thus creating a trunk vasculature that consists almost exclusively of aISV, rescued hypersprouting in *flt1* mutants (Fig. 5c-e). Conversely, promoting vISV numbers augmented vascular branching in *flt1* mutants (new Fig. 5f,g). Based on these data we conclude that, when arterial derived endothelial cells are integrated into vISV and remain in the dorsal part of vISV, they adapt the local sprouting phenotype.

The outstanding question is: which phenotypical adaptation occurs that renders (artery derived) endothelial cells in the venous domain responsive to loss of *flt1*. One signaling pathway that particularly differs between arteries and veins is Notch. In the zebrafish trunk vasculature, Notch promotes arterial identity, and Notch is a repressor of sprouting and active in arteries, not in veins (Quillien et al. 2014). The group of Lawson (Quillien et al. 2014) has convincingly shown that upon AV remodeling, Notch signaling is restricted to arteries and absent in veins. In the next section we provide evidence showing that Notch restricts arterial endothelium from sprouting upon loss of *flt1*. We thus postulate that artery derived endothelial cells, when integrated into the venous domain, lose Notch, and lose the repressive actions exerted by Notch, enabling them to respond to changes in Vegfaa bio-availability upon loss of *flt1* (and also in *vhl* mutants).

Comment: point 3, if Vegfaa and sFlt1 from the neurons affect the ECs of the secondary sprouts, the cells constituting DLAV must be changed from the pre-existing arterial ECs to mixed population of ECs consisting of the pre-existing ECs and migrating ECs of secondary sprouts. In the *Flt1* mutant embryos, the number of ECs of DLAV must be increased in addition to hyper-branching of vISV. In Figure 3l and 3l', the cell number of vISV was counted. Similarly, the cell number of DLAV should be counted. Ideally, the number of Flt4-positive ECs originating from the secondary sprouts in DLAV should be counted. Indeed, the width of DLAV in wild type embryos appears to be less than that of *flt1^{ka601}* mutant (Figure 2e' and 2g').

Response: in line with the reviewers' suggestion we quantified the number of endothelial cells in the aISV, vISV, DLAV, DA, and PCV of WT and *flt1^{ka601}* mutants (Fig. 3k-m). We find that in *flt1^{ka601}*, endothelial cell numbers were increased in all these domains. This is consistent with Vegfaa mediated endothelial proliferation. We did not find significant changes in DLAV width (Fig. 3n). With respect to PCV derived cells entering the DLAV, or PCV derived cells pushing arterial endothelium into the DLAV: for detailed response on the procedure we refer to our previous section on cell tracking.

We monitored migration of PCV/vISV derived venous cells (Supplementary movie 5) and observed them entering the DLAV. The reviewer is correct, we indeed find that the DLAV consists of mixed artery and vein derived endothelium. If we understand the reviewer correctly, she/he in addition wants to know what happens with artery derived cells when they are pushed out of the venous domain and into the DLAV, or with the venous derived endothelium when they enter the DLAV; can they give rise to sprouts upon loss of *flt1*. To adequately answer this question, it is essential to define the border of the venous domain at the vISV-DLAV interface, and to establish where the arterial domain ends in the DLAV. The DLAV is artery derived, and Notch appears active in this domain (Leslie et al. 2007; Quillien et al. 2014). Hence, if PCV derived venous endothelial cells move far enough into

the DLAV, they may acquire “arterial” characteristics including active Notch, restricting them from sprouting. The same holds for artery-derived endothelium when pushed out of the vISV into the DLAV. We only occasionally/rarely observed ectopic sprouting from the DLAV upon loss of *flt1*. However, if we knock down *dll4* in *flt1* mutants, we observed branching of the DLAV (Fig. 5f,g), suggesting that Notch represses sprouting of endothelial cells in the DLAV. Arterial Notch is not affected by loss of *flt1*.

Ectopic sprouting and differences between arteries and veins; Notch and pericytes.

Comment: point 1. It is still unclear why only endothelial cells (ECs) expressing Flt4 (those in the secondary sprouts from the PCV) respond to Vegfaa from neural tube. The requirement of *Kdr1* for hyper-branching in *flt1^{ka601}* and/or *vhlhu2114* is clearly shown in the present study; however, it remains elusive why ECs of aISV do not respond to Vegfaa, although the ECs of aISV do express *Kdr1*. The ECs in vISV express highly sensitive *Kdr1* and might potentially respond to Vegfaa. As the authors describe in the title, why do only Flt4- plus *Kdr1*-expressing ECs instead of those expressing only *Kdr1* have high angiogenic potential?

Response: in 5 *vegfaa* gain of function scenarios we observed ectopic venous sprouting. Ectopic arterial sprouting was observed in cases where we augment *vegfaa* gain of function by combining mutants (*flt1;vhl* and *ptena;ptenb*) or overexpressed *vegfaa* under control of a neuronal promoter (Fig. 5b). Thus both arteries and veins can make ectopic sprouts in response to Vegfaa, but arteries seem to require more Vegfaa than veins (Fig. 5b). One possible explanation for these results is the existence of an artery specific signaling pathway, repressing sprouting. We find that Notch may play a more prominent role than we originally reported.

It is well established that Notch acts as a negative regulator of sprouting angiogenesis, and Notch signaling is high in arteries and low in veins. We tried to inhibit Notch in arteries using an aISV specific UAS driven *dnMAML* approach. In line with Notch acting as a repressor, we find that inhibiting Notch with *dnMAML* using the UAS driven approach resulted in arteries forming ectopic sprouts, albeit at levels lower than observed in veins. Based on feedback from colleagues in the zebrafish community and a very recent paper published by the group of Lawson (Shin et al. 2016), our technical approach may have a technical limitation. These authors report that UAS driven approaches, result in mosaic and variable expression of the target gene. If this is correct, we cannot rule out that we did not achieve complete Notch inhibition with our *dnMAML* approach. Residual Notch activity may have contributed to repressing sprouting – thus yielding lower sprouting levels in arteries when compared to veins. To overcome this technical problem we now used a pharmacological approach to inhibit Notch (Stegmaier et al. 2014). We added the Notch inhibitor LY-411575 at 2dpf (after the AV remodeling) to *flt1^{ka601}* mutants and observed ectopic arterial sprouting, at levels twice as high as observed with the *dnMAML* approach (Fig. 6e,f). The difference between LY-411575 and *dnMAML* indeed supports that there was residual Notch activity in the latter scenario.

Reviewer 1 pointed us toward investigating endothelial cell numbers in detail, and asked what happens when we correct the observed events for the number of cells. We substantiated the endothelial cell data and show that upon loss of *flt1* both arterial and venous cell numbers were increased (Fig. 3k-m). We observed a clear difference between cell numbers in aISV and vISVs; in *flt1* mutants, aISVs on average contained 10 and vISV

15 endothelial cells. Thus, theoretically, if all endothelial cells in aISV (n=10) and vISV (n=15) would give rise to a sprout, one would still observe an AV difference (10 versus 15); the #vISV sprouts / # aISV sprouts ratio would be 1.5.

Of note: we complemented our study with knockdown of *dll4* in *flt1*^{ka601}. Loss of *dll4* prior to AV remodeling, results in a trunk vasculature that consists almost exclusively of venous ISVs (Fig. 5f,g) as previously described (Leslie et al. 2007). This is attributed to Dll4-Notch's role in specification of arterial identity, and loss of Notch induces venous identity. Accordingly, in support of a role for venous endothelium, loss of *dll4* in *flt1* mutants augmented branching when compared to *flt1* mutants (Fig. 5f,g). Of course Notch is tightly coupled to both vessel identity and sprouting (Leslie et al. 2007; Quillien et al. 2014). Veins show low Notch and thus lack a repressor, which may favor vISV sprouting. We added these new data to our manuscript and adapted our conclusions accordingly.

Comment: point 1, The coverage of aISV by mural cells must inhibit the sprouting from aISV. This might account for the difference of abnormal sprouts from aISV and vISV.

Response: We agree with the reviewer that coverage by mural cells may affect sprouting behavior; a vessel completely covered by mural cells will hardly sprout. Indeed AV differences exist in both the emergence of mural cells, and the mural cell coverage. In general arterioles are covered earlier than venules. However, when examining the literature, there is not a clear consensus on which markers specifically mark mural cells and differentiate between pericytes and smooth muscle. In zebrafish, several transgenic models exist to image mural cells *in vivo*. The group of Sarah Childs developed a transgenic line based on *acta2*; alpha-smooth muscle actin, a marker that, according to this group, marks both pericytes and smooth muscle (Whitesell et al. 2014). Analysis of the trunk vasculature using this transgenic line, *Tg(acta2:EGFP)*, showed that mural cells are scarce; the earliest time-point at which vascular mural cells were detected was at 96hpf, in the ventral portion of the dorsal aorta (Whitesell et al, 2014). Mural cells in ISVs were not observed at this time-point. Similar observations were made with antibodies against mural cell marker transgelin (aka Sm22-alpha-b) (Santoro et al. 2009)). Thus in these studies mural cells arrive well after we observe the emergence of ectopic sprouts (52hpf).

More recently the group of Naoki Mochizuki developed a different transgenic line to visualize mural cells, which is based on the pdgf receptor b promoter (Ando et al. 2016). *Pdgfr-b* is expressed by pericytes. Using this transgenic line, they reported that at 120hpf, aISV showed more pericytes than vISV (Ando et al. 2016). They furthermore noted that the pericytes adhered to the *flt1* positive arterial derived endothelium within the vISV (Ando et al. 2016). In time-lapse movies they noted emergence of the first pericytes around 58hpf near the dorsal aorta. Given the relatively early expression of this marker (when compared to *acta2* and transgelin), we decided to use this line. We investigated pericyte recruitment in the time-period 50-60hpf, the period during which we observe the emergence of ectopic sprouts. The data are presented in new Fig. 6g-i. Overall, pericytes were scarce in the trunk vasculature at this stage and most ISVs (about 88%) were not covered by pericytes. We find that at this developmental stage, only 9.94% of all aISV, and 1.91% of all vISV had pericytes (Fig. 6i). When comparing the dorsal part of the ISV, 2.48% of all aISV and 1.91% of all vISV displayed pericyte recruitment (Fig. 6i). While acknowledging the AV differences in pericyte recruitment, we believe that these pericyte numbers are too low to explain the AV sprouting differences.

Comment: point 4, Figure 1c and supplementary movie 1 clearly show the indenting of not only vISV but also aISV, suggesting that Vegfaa and sFlt1 might determine the route of extension of primary sprouts. However, the primary sprouts from the dorsal aorta and the formation of DLAV were unaffected in the Flt1 mutant fish. Why does this happen?

Response: first: Notch is active in developing aISVs and Notch restricts arteries from sprouting. Second: another factor that needs to be taken into consideration relates to timing of Vegfaa elevations as became evident from analyzing the *vhl* mutants (loss of *vhl* causes increased HiF-1a activity promoting *vegfaa* expression). In *vhl* mutants ectopic venous sprouts arise at the same time as in *flt1* mutants. In *vhl* mutants we show that neuronal *vegfaa* levels rise after formation of the aISV, and AV remodeling, and primary aISV sprouting is not affected (Supplementary Fig. 6b-d); ectopic venous sprouts form after 2.5dpf thus after the increase of *vegfaa*. This leaves open the option that upon loss of *flt1*, the increase in neuronal Vegfaa bio-availability during early stages, prior to AV remodeling is not sufficiently high to promote sprouting of aISVs with high Notch signaling. In line with this, we find that primary aISV development is not affected in our other gain of function scenarios (Supplementary Fig. 6b-d). However increased aISV endothelial cell numbers suggest that aISV do sense more Vegfaa upon loss of *flt1*; they just do not sprout. To make arteries sprout you need higher Vegfaa levels (by combining mutants), as depicted in Fig. 5b.

Comment: The location of ISVs and neural tube (like Figure 1c) at early time points (initial blood vessel formation from primary sprouts to formation of DLAV before the connection to the secondary sprout) in both wild type and Flt1 mutants should be analyzed carefully to examine the effects of Vegfaa and sFlt1 from neural tube on migration of ECs of primary sprouts. If the authors find intending of ISV of primary sprouts, it is unclear why only ECs of vISV were affected in Flt1 mutants (hyper-branched ECs are only in vISVs)? Because this point might puzzle the readers, the authors need to demonstrate the cause of difference of the effect of neural tube-derived molecules on primary sprouts and secondary sprouts.

Response: In line with the reviewers' request we in addition present new images on the location of the neural tube with respect to the positioning of the expanding aISV sprouts in WT and *flt1* mutants (Supplementary Fig. 6e-e'''). We show aISV in close proximity to the neural tube (Supplementary Fig. 6e'',e'''). We also carefully monitored aISV sprout expansion, filopodia length and numbers in WT, *flt1*^{ka601}, *vhl*^{hu2114}, and *flt1*;*vhl* double mutants at 24-30hpf and observed no significant changes (Supplementary Fig. 6b-d). As stated before, we find that Notch actively restricts sprouting in the arteries; thus when Notch is active, it robustly protects arteries from sprouting in physiologically relevant *vegfaa* gain of function scenarios.

Increased *vegfaa* without changes in arterial sprouting, has also been reported by others (Stahlhut et al. 2012). They report that loss of miR-1/miR-209, increased *vegfaa* resulting in proliferation of arterial endothelial cells in developing aISV during the period 24-30hpf; ectopic arterial sprouting or branching defects were not observed in this *vegfaa* gain of function scenario.

The reviewer furthermore asked if Vegfaa and sFlt1 produced by neurons could affect angiogenic behavior of developing aISV. To address this we overexpressed *sflt1* and *vegfaa* under control of a neuronal promoter during aISV development. We show that constitutive neuronal overexpression of *sflt1* inhibits the sprouting of aISVs (Supplementary Fig. 8k) consistent with sFlt1 scavenging the Vegfaa that is required for aISV development. Neuronal *vegfaa* gain of function resulted in thickened aISVs and a completely abnormal architecture of the trunk vasculature (Supplementary Fig. 8j; expression induced at 30hpf; GOF at earlier stages completely disrupts the trunk vasculature). We thus conclude that neuronal sFlt1 and Vegfaa derived can reach aISV and affect their development, provided that their levels are sufficiently high.

Taken together: we believe that there are qualitative and quantitative differences between arteries and veins in response to *vegfaa* gain of function. In arteries, active Notch robustly restricts sprouting in response to physiological elevations in Vegfaa (as in *flt1* mutant, *vhl* mutant, or loss of miR-1 & miR-206). However, arteries do sense increased Vegfaa levels as shown by increased cell numbers. To obtain arterial sprouting, one either has to inactivate Notch, or to increase Vegfaa levels to supra-physiological levels. The latter can be achieved by combining mutants, or by forced overexpression of neuronal *vegfaa*. In the venous domain, endothelial cells lack repressive signaling by Notch, which enables veins to respond to smaller elevations in Vegfaa when compared to arteries.

Comments: If the authors find the primary ISVs indenting neural tube, how is neuro-vascular interfaces of primary ISVs; the lateral interface between neural tube and ISVs and the dorsal interface between neural tube and DLAV, are regulated?

Response: we show that the DLAV can make additional branches; knock-down of *dll4* in *flt1* mutants augmented DLAV branching (Fig. 5f,g). This shows that in this (loss of Notch) scenario there is no physical hindrance that precludes diffusion of neuron-derived ligands toward the DLAV. However it could be that there are subtle differences in the neuro-vascular interface between aISV, vISV, DLAV and neural tube, specifically affecting diffusion toward aISV or vessels with high Notch signaling. Since the arteries arrive prior to veins, it suggests that arteries actively promote formation of such a barrier that is removed once they are remodeled into veins, pending the vascular Notch signaling status. We therefore performed transmission electron microscopy (data not shown). At the neuro-vascular interface, we observed that the neural tube is surrounded by a basal lamina that may affect distribution of the neural derived molecules. To address AV differences in bio-availability of Vegfaa in this area we would need gold-immuno labeled antibodies that can label aISV, vISV, ECM compounds, Vegfaa and sFlt1 on TEM sections. Given the fact that we present new experimental data explaining the AV differences in ectopic sprouting, we feel that a comprehensive analysis of the composition of the neuro-vascular interface and Vegfaa bio-availability at the neuro-vascular interface in aISV and vISV at ultrastructural level is better suited for another manuscript.

Comment: If the points #1-#4 are addressed, the main message of this study might be changed. Vegfaa and sFlt1 might regulate not only the spatial patterning of vISV but also the DLAV formation. Accordingly, Figure 7 needs to be revised.

Response: we would like to thank the reviewer for the stringent analysis of our data, and for pointing us toward performing the cell tracking experiments. Based on the new data sets we adapted our conclusion, title, and relevant parts in the discussion. We revised figure 7 (new Fig. 9).

Comment: (1) Purple is not appropriate for the line. Other type of white broken line should be used in Figure 1e. (2) 30hpf in Figure 1 and 26(space)hpf in Figure 2. This expression should be used with consistency.

Response: we changed the color for the lines in Fig. 1. We adapted the indication of developmental stage (without space) according to the reviewers' suggestion.

Reviewer #4.

Comment: in their manuscript "Neuronal Flt1 controls spinal cord vascularization involving venularized arterial endothelium with high angiogenic potential." Wild and colleagues aim to study the molecular cross-talk between developing tissue and the endothelium. By using Zebrafish as an in vivo model they claim that developing spinal cord neurons coordinate endothelial cells proliferation at the neurovascular interface by titrating/buffering the local bioavailability of neuron-derived Vegfaa through expression of sFlt1. Mechanistically, they claim that veins at the neurovascular interface have higher expression of Kdr1 receptors enabling them to have a greater sprouting potential in response to Vegfaa than arteries. It is proposed that such mechanism allows neurons to fine-tune neuro-vascular development. General comment: This paper aims to characterize the formation of organ-specific vasculature by studying the cross-talk between developing tissue and the endothelium. A combination of both in-vivo and in-vitro assays is used to characterize the molecular mechanisms on how neuro-derived sFlt1 controls the bioavailability of Vegfaa to regulate angiogenesis at the neurovascular interface. The quality and depth of analysis of the in-vivo data is remarkable. On the other hand, the mechanistic in-vitro data is not so compelling and needs further validation. Specifically, the claim that venularized arterial endothelial cells have a unique angiogenic potential due to a higher expression of the Kdr1 receptor remains elusive. Overall, the manuscript is nicely written and presented, uses state of the art genetic techniques but tackles a very well characterized biological phenomenon (VEGF-sFLT1 antagonism). The conceptual novelty is, thus, somewhat limited.

Response: we would like to thank reviewer 4 for the constructive comments; the reviewers' remark that "the quality and depth of analysis of the in-vivo data is remarkable" is well taken. In line with the reviewers' suggestion, we present new data sets that substantiate the contribution of neuronal *flt1* and we addressed the mechanism contributing to the AV sprouting difference. In short: 1) we compared *flt1* expression with neural guidance gene expression and find that they are expressed in a comparable range (Supplementary Fig. 1). 2) To further support the real-time PCR data and the neuronal expression observed in the *TgBAC(flt1:YFP)* reporter we performed cell transplantation experiments and showed that transplantation of *flt1* mutant neuronal cells into WT hosts induced ectopic venous sprouting. 3) To substantiate the involvement of neuronal *sflt1* at the genetic level, we specifically targeted neuronal *sflt1* and found ectopic venous sprouting. 4) We provide evidence showing that Notch is what causes the AV differences in ectopic sprouting upon loss of *flt1*. Taken together we feel that our data establish a functional role for neuronal sFlt1 in controlling sprouting at the neuro-vascular interface. In line with the reviewers' suggestion, we adapted our title.

Comment: Figure 1 (h) & (i)- Regarding the quantification of the mRNA expression levels, it is not clear how this data is presented. Does F.C. means Fold-change? The relative mRNA expression is relative to which sample?

Response: We apologize for this unclarity. It should be: relative expression of indicated gene, normalized to *mflt1* (set at 1.0). The mRNA levels presented in the graph indicate the "expression of the gene of interest" normalized to *mflt1*. A value of 2 means that this gene is expressed 2 fold higher when compared to *mflt1* in this cell population. We adapted the legends and y-axis of these panels accordingly.

Comment: mRNA quantification by qPCR for a neuronal specific gene (when comparing GFP positive vs. GFP-negative cells) should also be included to demonstrate the purity of the sorted GFP-positive population. Additionally, authors should present data as in Figure 5(b): mRNA expression levels of these genes should be compared to non-neuronal cells (GFP-negative cells).

Response: in line with the reviewers' suggestions we measured the expression of the neuron specific, pan-neuronal marker HuC in GFP positive and GFP negative cells, sorted from *Tg(huc:egfp)* and *Tg(mnx1:egfp)* transgenic embryos respectively (Supplementary Fig. 1a,b). HuC is specific for neurons, and expressed in all neuronal cell populations. The GFP negative population sorted from *Tg(huc:egfp)* reflects the non-neuronal cell population. In addition we also include a comparison of *sflt1*, *mflt1* expression with the expression of neural guidance genes (Supplementary Fig. 1a'',b''), and find that they are expressed at comparable or higher levels (Supplementary Fig. 1a'',b''). To demonstrate the physiological relevance of neuronal *flt1* mRNA expression level we used several genetic approaches and added a neuronal cell transplantation experiment. We present five lines of evidence supporting a physiologically relevant role for neuronal *flt1* in angiogenesis (see below).

Of note: the *huc* promoter is active in all neurons; in *Tg(huc:egfp)* transgenic embryos, sensory nerves, motoneurons, and interneurons, all express GFP. *Mnx1* marks only motoneurons, and in *Tg(mnx1:gfp)* transgenic embryos, only motoneurons express GFP; other neuronal cell populations like sensory neurons, and interneurons are GFP negative. Thus upon sorting, the population of GFP negative (GFP-) cells sorted from *Tg(mnx1:gfp)* embryos includes sensory neurons, interneurons, as well as non neuronal cell populations including vascular endothelium. This GFP negative population will thus express neuronal markers because it contains sensory neurons, and interneurons. In contrast, when we sort from the pan-neuronal marker *Tg(huc:egfp)* transgenic line (labeling all neurons), the GFP negative population will only include non-neuronal cells. In line with this: when we compared *huc* expression in *Mnx-GFP+* versus *Mnx-GFP-* cells (Supplementary Fig. 1b) with *huc* expression in *Huc-GFP+* versus *Huc-GFP-* cells (Supplementary Fig. 1a), we observed higher expression in cells sorted from the *Tg(huc:egfp)* when compared to *Tg(mnx1:gfp)*. This issue is not related to the purity of the sorted cells but related to differences in the (neuronal) cell populations that are included in GFP+ and GFP- fraction.

We do agree with the reviewer that in order to obtain almost 100% purity, at least three successive FAC sorting cycles would be needed. Given the extreme loss of mRNA occurring during each cycle, several thousands of embryos would be required. Therefore, to firmly establish that neuronal *flt1* is indeed physiologically relevant we generated additional data. First: we transplanted *flt1* mutant neuronal cells into WT hosts and observed ectopic venous sprouting (Fig. 8g-i). Second: we genetically targeted neuronal *sflt1* and observed sprouting (Supplementary Fig. 8b,c). In addition we generated neuronal specific *flt1* mutants with CRISPR/Cas9 and provide a neuron specific *sflt1* rescue in *flt1* mutants (Fig. 8a-f). In WT we show that loss of neuronal *sflt1* parallels with onset of spinal cord vascularization during later stages (Fig. 7g-i and Supplementary Fig. 7e-g). Taken together we feel that these data support the concept that neurons produce physiologically relevant sFlt1 levels regulating angiogenesis at the neuro-vascular interface.

Comment: The author's claim that "two different neuronal reporter lines showed expression of *mflt1*, *sflt1*, *kdr1*, *kdr*, *flt4* and the ligands *vegfaa*, *vegfab*, and *plgf*". When comparing the results from the two neuronal reporters used, the relative mRNA expression of *sflt1* is shown to be down-regulated in HuC+ neurons and up-regulated in *mnx1*+ neurons. The same goes for other genes analyzed. What is the biological relevance of this finding? Could this be related to the purity of the sorted population?

Response: Indeed, in both reporter lines we detected *mflt1*, *sflt1*, and *vegf* ligand mRNA expression in the GFP+ fraction of the FAC-sorted cells. We agree that these graphs may give the visual impression that genes are down-regulated. However, the individual graphs cannot be compared without taking into consideration the differences in cell populations that are included in the GFP+ and GFP- fractions (see also response to previous comment). As described above, in Huc transgenic the GFP+ fraction contains sensory, motor, and interneurons, whereas in *Mnx1* transgenic, the GFP+ fraction contains only motor neurons. Theoretically, if motoneurons are the only cell population that expresses *flt1*, their contribution will be "diluted" once mixed with sensory and interneuron populations, as occurs in the Huc-GFP+ sorted population. Thus the biological relevance of the observed differences in *flt1* most likely reflects heterogeneity in *flt1* expression and *flt1* splicing among the different neuronal populations.

The statement that we want to make is that mRNA analysis of FAC-sorted cells derived from neuronal reporter lines supports *mflt1* and *sflt1* expression by neurons. This is in support of neuronal *flt1* expression observed in the *TgBAC(flt1:YFP)* reporter (active *flt1* promoter activity in neurons, Fig. 1a,b). And to prove that these *flt1* levels are indeed of physiological relevance, we performed genetic experiments (CRISPR/Cas9 neuron specific *flt1* mutant; neuron specific *sflt1* LOF embryo; and *flt1* mutant neuronal cell transplantation experiment), all supporting the concept that neuronal *flt1* restricts sprouting at the neuro-vascular interface.

Comment: Figure 2 - The author's claim that "The vascular phenotype observed in the *flt1ka601* mutants thus most likely involved soluble Flt1." - Why did the authors not generate a specific sFlt1 mutant by targeting exon E11a? This appears to be the most direct way to support author's findings.

Response: to further strengthen the claim that neuronal *sflt1* is the cause of hyper-branching we generated neuronal tissue specific *sflt1* loss of function embryos (Supplementary Fig. 8b,c). For generating neuron specific *sflt1* loss of function embryos we used a recently developed miRNA based technique, with high knockdown efficiency, optimized for *in vivo* use in zebrafish (Giacomotto et al. 2015). To this end we employed multiple custom designed miRNAs directed against *sflt1* arranged in series with a common miR-155 backbone (see materials and methods part "Generation of tissue-specific miR155-*flt1*-1-2-3 knockdown constructs" and (Giacomotto et al. 2015)). To obtain tissue specificity we cloned the expression constructs under control of vascular (*flt1^{enh}*) and neuronal (*Xla.Tubb*) specific promoters. We observed that targeting neuronal *sflt1* resulted in ectopic venous sprouting (Supplementary Fig. 8b,c). Targeting vascular *sflt1* did not induce hypersprouting (Supplementary Fig. 8b,d).

In addition we tried to generate a stable *sflt1* mutant by targeting exon11a using a

CRISPR/Cas9 approach. We used multiple sgRNAs directed against the exon-intron boundary, which according to prediction algorithms should inhibit splicing. However, RNA analysis subsequently showed that, while the sgRNAs worked and generated the expected indels, splicing still occurred due to emergence of several cryptic splice sites in the upstream intronic region. Removing all these potential cryptic splice sites requires a large deletion, which is technically challenging with the currently available techniques. We are working on optimizing, but feel that for the revision, our neuronal specific targeting of *sflt1* using the multiplexed miRNA approach, and neuron specific *sflt1* rescue, offers an elegant alternative.

Comment: Figure 4(h)- The author's claim that "Intersegmental veins express more *Kdr1* receptor than arteries". The method used to support such finding is indirect and the small differences reported may not have biological relevance. Thus, more experimental evidence is needed to support the concept that veins at the neurovascular interface have a greater sprouting potential in response to Vegfaa than arteries due to higher expression of *Kdr1* receptors.

Response: we agree with the reviewer that our statement on *Kdr1* receptors was over enthusiastic. Based on feedback from the reviewers, the requested analyses and new data sets, we removed this phrase. We now present new data addressing the cause of the arterial-venous sprouting differences and reconsidered other explanations for our observations. In short: we find that Notch may play a more prominent role than originally anticipated.

It is well established that Notch acts as a negative regulator of sprouting angiogenesis, and Notch signaling is high in arteries and low in veins. We tried to inhibit Notch in arteries using an aISV specific UAS driven *dnMAML* approach. In line with Notch acting as a repressor, we find that inhibiting Notch with *dnMAML* using the UAS driven approach resulted in arteries forming ectopic sprouts, albeit at levels lower than observed in veins. Based on feedback from colleagues in the zebrafish community and a very recent paper published by the group of Lawson (Shin et al. 2016), our technical approach may have a technical limitation. These authors report that UAS driven approaches, result in mosaic and variable expression of the target gene. If this is correct, we cannot rule out that we did not achieve complete Notch inhibition with our *dnMAML* approach. Residual Notch activity may have contributed to repressing sprouting – thus yielding lower sprouting levels in arteries when compared to veins. To overcome this technical problem we now used a pharmacological approach to inhibit Notch (Stegmaier et al. 2014). We added the Notch inhibitor LY-411575 at 2dpf (after the AV remodeling) to *fll1^{ka601}* mutants and observed ectopic arterial sprouting, at levels twice as high as observed with the *dnMAML* approach (Fig. 6c-f). The difference between LY-411575 and *dnMAML* indeed supports that there was residual Notch activity in the latter scenario.

Reviewer 1 pointed us toward investigating endothelial cell numbers in detail, and asked what happens when we correct the observed events for the number of cells. We substantiated the endothelial cell data (Fig. 3k-m). We observed a clear difference between cell numbers in aISV and vISVs; in *fll1* mutants, aISVs on average contained 10 and vISV 15 endothelial cells.

Of note: we complemented our study with knockdown of *dll4* in *fll1^{ka601}*. Loss of *dll4* prior to AV remodeling, results in a trunk vasculature that consists almost exclusively of venous ISVs (Fig. 5f,g) as previously described (Leslie et al. 2007). This is attributed to Dll4-

Notch's role in specification of arterial identity, and loss of Notch induces venous identity. Accordingly, in support of a role for venous endothelium, loss of *dll4* in *flt1* mutants augmented branching when compared to *flt1* mutants (Fig. 5f,g). Of course Notch is tightly coupled to both vessel identity and sprouting (Leslie et al. 2007; Quillien et al. 2014). Veins show low Notch and thus lack a repressor, which may favor vISV sprouting upon loss of *flt1*. We added these new data to our manuscript and adapted our conclusions accordingly.

Comment: The title of the manuscript should emphasize soluble Flt1 as the major driver for the proposed mechanism.

Response: We adapted the title in line with the reviewers' suggestion.

Comment: Figure 1 legend - incorrect figure identification. (k) should be replaced by (j).

Response: we moved these panels to Supplementary Figure 1 and corrected the legend accordingly.

Comment: Figure 3 (m) - Tdtomoto should be replaced by tdtomato. Also, the concentration of MO that was used in this experiment is not clear. Please include this information in the figure legend.

Response: We corrected the spelling mistake. The morpholino dosage was 1ng; we added this information to the figure legends.

Bibliography

- Ando, K. et al., 2016. Clarification of mural cell coverage of vascular endothelial cells by live imaging of zebrafish. *Development (Cambridge, England)*, 143(8), pp.1328–39.
- Giacomotto, J., Rinkwitz, S. & Becker, T.S., 2015. Effective heritable gene knockdown in zebrafish using synthetic microRNAs. *Nature Communications*, 6(May), p.7378.
- Gordon, K. et al., 2013. Mutation in Vascular Endothelial Growth Factor-C, a Ligand for Vascular Endothelial Growth Factor Receptor-3, Is Associated With Autosomal Dominant Milroy-Like Primary Lymphedema Novelty and Significance. *Circulation Research*, 112(6).
- Hogan et al., 2009. Vegfc/Flt4 signalling is suppressed by Dll4 in developing zebrafish intersegmental arteries. *Development (Cambridge, England)*, 136(23), pp.4001–9.
- Krueger, J. et al., 2011. Flt1 acts as a negative regulator of tip cell formation and branching morphogenesis in the zebrafish embryo. *Development (Cambridge, England)*, 138(10), pp.2111–20.
- Leslie, J.D. et al., 2007. Endothelial signalling by the Notch ligand Delta-like 4 restricts angiogenesis. *Development*, 134(5), pp.839–844.
- Mukouyama, Y., 2008. Neuro-vascular interaction during mouse development. *The FASEB Journal*, 22, p.391.1.
- Mukouyama, Y.S. et al., 2002. Sensory nerves determine the pattern of arterial differentiation and blood vessel branching in the skin. *Cell*, 109(6), pp.693–705.
- Nugent, J. & O'Connor, M., 1983. Ciba Foundation Symposium 100 - Development of the Vascular System. , pp.95–100.
- Quillien, A. et al., 2014. Distinct Notch signaling outputs pattern the developing arterial system. *Development*, 141(7), pp.1544–1552.
- Risau, W., 1995. Differentiation of endothelium. *FASEB journal: official publication of the Federation of American Societies for Experimental Biology*, 9(10), pp.926–33.
- Rocha, S.F. et al., 2014. Esm1 modulates endothelial tip cell behavior and vascular permeability by enhancing VEGF bioavailability. *Circulation research*, 115(6), pp.581–90.
- Ruhrberg, C. & Bautch, V.L., 2013. Neurovascular development and links to disease. *Cellular and molecular life sciences: CMLS*, 70(10), pp.1675–84.
- Santoro, M.M., Pesce, G. & Stainier, D.Y., 2009. Characterization of vascular mural cells during zebrafish development. *Mechanisms of development*, 126(8–9), pp.638–49.
- Shin, M. et al., 2016. Vegfa signals through ERK to promote angiogenesis, but not artery differentiation. *Development (Cambridge, England)*, p.dev.137919.
- Stahlhut, C. et al., 2012. miR-1 and miR-206 regulate angiogenesis by modulating VegfA expression in zebrafish. *Development (Cambridge, England)*, 139(23), pp.4356–64.
- Stegmaier, J. et al., 2014. Automated prior knowledge-based quantification of neuronal patterns in the spinal cord of zebrafish. *Bioinformatics*, 30(5), pp.726–733.
- Strasser, G.A., Kaminker, J.S. & Tessier-Lavigne, M., 2010. Microarray analysis of retinal endothelial tip cells identifies CXCR4 as a mediator of tip cell morphology and branching. *Blood*, 115(24), pp.5102–10.
- del Toro, R. et al., 2010. Identification and functional analysis of endothelial tip cell-enriched genes. *Blood*, 116(19), pp.4025–33.
- Whitesell, T.R. et al., 2014. An α -smooth muscle actin (acta2/asma) zebrafish transgenic line marking vascular mural cells and visceral smooth muscle cells. *PLoS one*, 9(3), p.e90590.

Reviewer #1 (Remarks to the Author)

-

Reviewer #2 (Remarks to the Author)

The authors have addressed all of my concerns. As a matter of fact, the revised version of the manuscript has been substantially improved from the previous version. By providing more elaborate lineage tracing and exhaustive cell transplantation experiments, the authors convincingly demonstrate the importance of neuronal sFlt1 in regulating Vegf-A gradient during spinal cord vascularization.

Reviewer #4 (Remarks to the Author)

Wild and colleagues provide an extensive revision of their manuscript that was already positively reviewed in the initial submission. The authors address all comments of this and the other reviewers in full and provide compelling evidence for a central role of neuronal sFlt1 and Vegfaa signalling in spinal cord vascularisation. The paper will be of interest to the broad readership of Nature Communications and should be published soon.

Point-by-point response to issues raised by reviewers:

Reviewer #1 (Remarks to the Author):

We thank reviewer 1 for supporting the publication of our manuscript.

Reviewer #2 (Remarks to the Author):

Remark: The authors have addressed all of my concerns. As a matter of fact, the revised version of the manuscript has been substantially improved from the previous version. By providing more elaborate lineage tracing and exhaustive cell transplantation experiments, the authors convincingly demonstrate the importance of neuronal sFlt1 in regulating Vegf-A gradient during spinal cord vascularization.

Response: We thank reviewer 2 for the kind feedback. The remark that the authors convincingly demonstrate the importance of neuronal sFlt1 is well taken.

Reviewer #4 (Remarks to the Author):

Remark: Wild and colleagues provide an extensive revision of their manuscript that was already positively reviewed in the initial submission. The authors address all comments of this and the other reviewers in full and provide compelling evidence for a central role of neuronal sFlt1 and Vegfaa signalling in spinal cord vascularisation. **The paper will be of interest to the broad readership of Nature Communications and should be published soon.**

Response: We thank reviewer 4 for the kind feedback.